

# Assessing the representation of the Australian carbon cycle in global vegetation models

Lina Teckentrup[1,2], Martin G. De Kauwe[1,2,3], Andrew J. Pitman[1,2], Daniel Goll[4], Vanessa Haverd[5,†], Atul K. Jain[6], Emilie Joetzjer[7], Etsushi Kato[8], Sebastian Lienert[9], Danica Lombardozzi[10], Patrick C. McGuire[11], Joe R. Melton[12], Julia E. M. S. Nabel[13], Julia Pongratz[13,14], Stephen Sitch[15], Anthony P. Walker[16], and Sönke Zaehle[17]

[1]ARC Centre of Excellence for Climate Extremes, Sydney, NSW, Australia
[2]Climate Change Research Centre, University of New South Wales, Sydney, NSW, Australia
[3]Evolution & Ecology Research Centre, University of New South Wales, Sydney, NSW 2052, Australia
[4]Université Paris Saclay, CEA-CNRS-UVSQ, LSCE/IPSL, Gif sur Yvette, France
[5]CSIRO Oceans and Atmosphere, G.P.O. Box 1700, Canberra, ACT 2601, Australia
[6]Department of Atmospheric Sciences, University of Illinois, Urbana, IL 61821, USA
[7]CNRM, Université de Toulouse, Météo-France, CNRS, Toulouse, France, Unite mixte de recherche 3589 Meteo-France/CNRS, 42 Avenue Gaspard Coriolis, 31100 Toulouse, France
[8]Institute of Applied Energy (IAE), Minato-ku, Tokyo 105-0003, Japan
[9]Climate and Environmental Physics, Physics Institute and Oeschger Centre for Climate Change Research, University of Bern, Bern, Switzerland
[10]National Center for Atmospheric Research, Climate and Global Dynamics, Terrestrial Sciences Section, Boulder, CO 80305, USA
[11]Department of Meteorology, Department of Geography & Environmental Science, National Centre for Atmospheric Science, University of Reading, Reading, UK
[12]Climate Research Division, Environment and Climate Change Canada, Victoria, BC, Canada
[13]Max Planck Institute for Meteorology, Hamburg, Germany
[14]Department of Geography, LMU, Munich, Germany
[15]College of Life and Environmental Sciences, University of Exeter, Exeter EX4 4RJ, UK
[16]Environmental Sciences Division and Climate Change Science Institute, Oak Ridge National Laboratory, Oak Ridge, Tennessee, USA
[17]Max Planck Institute for Biogeochemistry, P.O. Box 600164, Hans-Knöll-Str. 10, 07745 Jena, Germany
[†]deceased, 19 January 2021

**Correspondence:** Lina Teckentrup (l.teckentrup@unsw.edu.au)

**Abstract.** Australia plays an important role in the global terrestrial carbon cycle on inter-annual timescales. While the Australian continent is included in global assessments of the carbon cycle such as the global carbon budget, the performance of dynamic global vegetation models (DGVMs) over Australia has rarely been evaluated. We assessed simulations of net biome production (NBP) and the carbon stored in vegetation between 1901 to 2018 from 13 DGVMs (TRENDY v8 ensemble). We focused our analysis on both Australia's short-term (inter-annual) and long-term (decadal to centennial) terrestrial carbon dynamics. The TRENDY models simulated differing magnitudes of NBP on inter-annual timescales, and these differences contributed to carbon accumulation in the vegetation on decadal to centennial timescales (-4.7–9.5 PgC). We compared the TRENDY ensemble to several satellite-derived datasets and showed that the spread in the models' simulated carbon storage resulted from varying changes in carbon residence time rather than differences in net carbon uptake. Differences in simulated





long-term accumulated NBP between models were mostly due to model responses to land-use change. The DGVMs also simulated different sensitivities to atmospheric carbon dioxide ($CO_2$) concentration, although notably, the models with nutrient cycles did not simulate the smallest NBP response to $CO_2$. Our results suggest that a change in the climate forcing did not have a large impact on the carbon cycle on long timescales. However, the inter-annual variability in precipitation drives the year-to-year variability in NBP. We analysed the impact of key modes of climate variability, including the El Niño Southern Oscillation

(ENSO) and the Indian Ocean Dipole (IOD) on NBP. While the DGVMs agreed on sign of the response of NBP to El Niño and La Niña, and to positive and negative IOD events, the magnitude of inter-annual variability in NBP differed strongly between models. In addition, we identified differences in the timing of simulated phenology and fire dynamics associated with differences in simulated/prescribed vegetation composition and process representation. Model disagreement in simulated vegetation carbon, phenology and apparent carbon residence time, indicates the models have different types of vegetation cover across

Australia (whether prescribed or emergent). Our study highlights the need to evaluate parameter assumptions and the key processes that drive vegetation dynamics, such as phenology, mortality and fire, in an Australian context to reduce uncertainty across models.

## 1 Introduction

Decadal variability in the growth rate of atmospheric carbon dioxide ($CO_2$) is strongly influenced by variability in the uptake

and release of carbon by the oceans and the terrestrial biosphere (Ballantyne et al., 2012; Raupach et al., 2008). The inter-annual variability (IAV) in the $CO_2$ growth rate is dominated by terrestrial processes (e.g. vegetation productivity, respiration and fire emissions) and their responses to both temperature and precipitation, driven by modes of climate variability (Ahlström et al., 2015; Zhang et al., 2018; Poulter et al., 2014). The El Niño Southern Oscillation (ENSO) is the dominant mode of global variability (e.g. Ahlström et al., 2015) and contributes around 26% to IAV in global gross primary production (GPP; Zhang

et al., 2019). Globally, ENSO has also been shown to explain more than 40% of satellite-derived net primary production (NPP) variability, mainly driven by the response of Southern Hemisphere ecosystems (Bastos et al., 2013) and in particular, semi-arid ecosystems (Zhang et al., 2018).

Around 70% of the land in Australia is classified as either arid or semi-arid (Brown et al., 2008). Oceania has been found to contribute significantly to the uncertainty in global and regional carbon budgets (Bastos et al., 2020) and the important role

played by semi-arid ecosystems in explaining the variability of the global carbon cycle was highlighted by the 2011 La Niña event. While on average, ∼ 17% of IAV in net biome production (NBP) was attributable to Australia over the historical period, during the 2011 La Niña event, Australia contributed around 60% to the global carbon sink (e.g. Poulter et al., 2014). A recent study suggested that the Australian terrestrial carbon sink may be enhanced due to more extreme wet-events projected for future decades (Ma et al., 2016). This carbon uptake enhancement has been associated with the asymmetric response of GPP

to precipitation, i.e. positive GPP anomalies tend to be larger than negative ones (based on upscaled global flux tower data; Haverd et al., 2017), in combination with vegetation expansion linked to rainfall (based on a single dynamic vegetation model;





Poulter et al., 2014). At the same time, a series of studies have also identified evidence of rising $CO_2$ leading to a marked greening of the Australian continent (Donohue et al., 2009, 2013; Ukkola et al., 2016b; Trancoso et al., 2017).

Dynamic global vegetation models (DGVMs) are commonly used to explore large–scale responses of the carbon cycle

to climate variability and climate change (Friedlingstein et al., 2019; Le Quéré et al., 2018). However, comparing NBP (the change in carbon stocks including carbon losses due to disturbance) between different DGVMs shows a large model spread with substantial variations of annual global NBP of up to 3 PgC yr$^{-1}$ when forced with the same meteorological drivers (Le Quéré et al., 2018). This demonstrates large uncertainties in the representation of the terrestrial carbon cycle and the associated response to climate and land-use change in DGVMs. Roxburgh et al. (2004) concluded from an Australian based evaluation of

NPP that DGVMs were unable to capture the interactions between terrestrial biosphere and atmosphere (models simulated a fivefold variation in annual NPP), citing weaknesses related to the role of water and nutrients in limiting productivity. A number of more recent studies have improved our understanding of the Australian terrestrial carbon cycle, employing flux tower data (e.g Tarin et al., 2020), satellite derived carbon fluxes (e.g. Cleverly et al., 2016) and regional biospheric modelling (e.g. Haverd et al., 2013, 2016b). Nevertheless, while the Australian continent is included in global assessments of the carbon cycle, DGVM

performance over Australia has rarely been evaluated. The need for such an evaluation is emphasized by recent studies that have used DGVM simulations to underline the importance of Australia's ecosystems to the inter-annual variability of the global carbon cycle (Poulter et al., 2014; Ahlström et al., 2015). DGVM studies have also previously highlighted vegetation responses to climate as critical to future projected changes in water resources (Ukkola et al., 2016a) and an enhanced future carbon sink (Kelley and Harrison, 2014) across Australia. Thus, evaluating DGVM skill is also equally important for assessing projected

changes in Australia's carbon and water cycles.

In this study, we assessed the terrestrial carbon cycle for Australia simulated by 13 DGVMs that are part of the TRENDY v8 ensemble. We examined drivers of both the long-term (decadal to centennial timescales) and short-term (inter-annual timescales) model responses of the carbon cycle. With these timescales in mind, we expected that model differences result from:

– sensitivity to increasing $CO_2$

– sensitivity to climate variability

– prescribed/ simulated land cover and interpretation of land-use change

– assumed/emergent functional representation of Australian vegetation

We therefore examined the contribution made by each of these differences across the TRENDY v8 ensemble. Our goal was

to isolate, to the extent possible, the causes of differences among the TRENDY v8 ensemble as a first step towards prioritising areas to focus on to resolve the differences among the models.



## 2    Materials and Methods

### 2.1    Models and simulations

We used the 13 dynamic vegetation models that are part of the TRENDY v8 model ensemble (Friedlingstein et al., 2019):
CABLE–POP, CLASS–CTEM, CLM5.0, ISAM, ISBA-CTRIP, JSBACH, JULES-ES, LPX-Bern, OCN, ORCHIDEE, ORCHIDEE-
CNP, SDGVM and VISIT. All models used identical forcing inputs and followed a common simulation protocol over the period
1700–2018 (see appendix figure B1 and description in the appendix for more detailed information). However, each model used
a model-specific interpretation of the land-use change forcing.

We remapped all model outputs and satellite datasets (see below) to a common 0.5° grid using first order conservative
regridding (except for the comparison with the NATT data). For this analysis, we defined a year to start in July and end in June
(except for the $C_{Veg}$ analysis which followed calendar years) to capture the Southern Hemisphere growing season and capture
full El Niño and La Niña events, which usually start and end in austral winter. We expressed the change in the variables as the
difference to the 1901–1930 average and processed the data with netCDF Operators (NCO; version 4.7.7. http://nco.sf.net) and
climate data operators (CDO; version 1.9.5. http://mpimet.mpg.de/cdo). The data analysis was conducted with Python version
85  3.

### 2.2    Satellite Data

To assess model simulations, we used several satellite-derived data sets. We note that direct continental-scale measurements of
the variables below do not exist, and that the satellite products also rely on models themselves.

*Gross primary production/phenology*

We compared the simulated the GPP phenology cycle to GPP derived from solar-induced chlorophyll fluorescence (SIF)
(GOSIF-GPP; Li and Xiao, 2019). To ensure the GOSIF-GPP phenology accurately captured phenology, we also compared
these data to a satellite-derived leaf-area index (LAI) product (modified Copernicus Service information, 2020; see appendix
figure B2). GOSIF-GPP is based on the OCO-2-based SIF product (GOSIF) and derived by assuming linear relationships
between SIF and GPP estimated from eddy covariances sites. Uncertainties were accounted for by using eight different SIF-
GPP relationships (derived universally and biome-specific, with and without intercept at both site and grid cell levels). The
0.05° dataset covers the period 2000 to 2018 at an 8-day temporal resolution, that we aggregated to a monthly time step.

*Carbon stored in vegetation*

Estimates of aboveground carbon biomass can be derived from satellite estimates of vegetation optical depth (VOD; related
to the aboveground vegetation's water content, density and biomass). We used aboveground carbon biomass to assess the carbon
stored in vegetation in the TRENDY models and note that the datasets are not directly comparable since the satellite product
does not account for belowground carbon. We used global estimates of the annual average aboveground biomass carbon for
1993–2012 with a 0.25° spatial resolution derived from a series of satellite passive microwave instruments (Version 1.0; Liu
et al., 2015). Liu et al. (2015) used an empirical approach to convert a harmonised time series of VOD to above-ground biomass
carbon.





*Fire-related $CO_2$ fluxes*


We used two different satellite estimates of fire $CO_2$ emissions: the Global Fire Emissions Database version 4 (GFED4s) described in van der Werf et al. (2017) as well as the Copernicus Atmosphere Monitoring Service Global Fire Assimilation System (CMAS-GFAS; Kaiser et al., 2012). GFED4s has a spatial resolution of 0.25° with a monthly time step and provides data from 1997–2020. We note that starting from 2017, the GFED4s data are flagged as a beta version, i.e. starting 2017, fire

emissions are derived from active fire detections instead of burned area. CAMS-GFAS covers the years 2003–2020 on a 0.1° horizontal grid. The source for both estimates is the MODIS satellite but different variables and methods are used to derive the fire $CO_2$ emissions (see e.g. van der Werf et al., 2017; Kaiser et al., 2012; Li et al., 2019; Pan et al., 2020, for more detail).

*Landcover fraction*

We used the MODIS/Terra Vegetation Continuous Fields Yearly L3 Global 250m SIN Grid V006 dataset for a comparison

of satellite derived and landcover in DGVMs. This product estimated the global landcover fraction using surface reflectance, brightness temperature and the MODIS Global 250m Land/Water Map (DiMiceli et al., 2017).

## 2.3 Fluxsites/ North Australian Tropical Transect sites

We analysed the TRENDY models in context of the North Australian Tropical Transect (NATT) gradient, which encompasses a precipitation gradient ranging from ∼720 to 1650 mm yr$^{-1}$. The NATT was established to understand the function of savanna

ecosystems, the predominant landscape in north Australia. We used the four sites Howard Springs (AU-How), Daly Uncleared (AU-DaS), Dry River (AU-Dry) and Sturt Plains (AU-Stp; see appendix table B1). We chose the simulations with transient $CO_2$ concentration, climate and land-use change. In order to compare the models to the flux sites, we isolated the corresponding gridcells based on the site coordinates and calculated the net ecosystem exchange (NEE) as a balance between GPP and terrestrial ecosystem respiration (TER). We further divided the data into the wet (November–April) and dry (May–October)

season.

Although we showed flux tower observations and simulations of NEE, GPP and TER together, we note that the spatial scales of observed ecosystem fluxes (eddy covariance) and simulations by a DGVM are not directly comparable given the flux tower footprint is ∼ 1 km$^2$ vs. ∼ 3000 km$^2$ for a gridcell in a 0.5° grid. Consequently, DGVMs cannot represent local features such as heterogeneous environmental conditions and land cover, or biogeographical feedbacks with atmospheric conditions specific

to the site (Piao et al., 2013; Luo, 2007). However, given the meteorology recorded at the sites and the meteorology used to drive the TRENDY simulations were highly correlated (see appendix figure B3), we assumed the TRENDY models simulated the vegetation in a similar climate as the observed. We also note that due to data loss, some of the observed data contained a high proportion of gap-filled data. This was particularly the case at Howard Springs site and for the TER flux across sites.

## 2.4 Vegetation classes

To group the analysis by regions, we classified Australia into six vegetation classes (tropics, savanna, warm temperate, cool temperate, mediterranean and desert; see appendix figure B4) following Haverd et al. (2012). These regions have distinct climate and biophysical characteristics and are based on the agro-climatic classification by Hutchinson et al. (2005).





## 2.5 Identification of modes of variability

Numerous studies highlighted the impact of different modes of climate variability on Australian weather patterns (e.g. Um-
menhofer et al., 2011). The El Niño Southern Oscillation (ENSO) and the Indian Ocean Dipole (IOD) drive variability in
precipitation, which in turn is the main driver of variability in the Australian carbon cycle (e.g Cleverly et al., 2016; Haverd
et al., 2013). We identified years corresponding to the phase of ENSO and IOD events (see appendix table B2). We used these
different climate modes to group our analysis of modelled carbon fluxes.

## 2.6 Apparent carbon residence time in vegetation

To understand model differences that are related to carbon stored in vegetation ($C_{Veg}$), we analysed the carbon residence time
in vegetation ($\tau$). We followed Friend et al. (2014) and Pugh et al. (2020) who defined the change in $C_{Veg}$ over time as

$$\frac{dC_{Veg}}{dt} = NPP - \frac{C_{Veg}}{\tau} \qquad (1)$$

Consequently, we calculated the change in the carbon residence time using annual timesteps according to

$$\tau = \frac{C_{Veg}}{NPP - \frac{dC_{Veg}}{dt}} \qquad (2)$$

## 2.7 PFT groups

Each of the TRENDY models has its own vegetation classification. To compare simulated/prescribed vegetation cover between
models, we grouped the model-specific plant functional types (PFTs) into ten PFT groups: evergreen trees ('EVG'), deciduous,
summergreen and raingreen trees ('DCD/ SMG/ RNG'), shrubs, savanna, C3 grass, C4 grass, C3 agriculture and C4 agriculture
(i.e. crops and pasture PFTs), bare ground and a group comprising the remaining landcover types, such as city, urban, or lakes.
For each model, we assigned the model-specific PFTs to these groups and finally compared the fraction of land covered by
each PFT group. Note that not all models account for all the vegetation groups, i.e. only four out of the ten PFT groups included
input from all of the models (EVG, DCD/ SMG/ RNG, C3 Grass, and C4 Grass).

## 3 Results

### 3.1 Net biome production and carbon stored in vegetation

Summed over Australia, the TRENDY model ensemble simulated large inter-annual variability in NBP (see fig. 1a) ranging
from -0.9 PgC yr$^{-1}$ to 1.4 PgC yr$^{-1}$. Figure 1a also shows a considerable model spread: the TRENDY models varied on
average by 0.5 PgC yr$^{-1}$ with a difference of up to 1.26 PgC yr$^{-1}$ between the most extreme models. Years with extremely
high or low NBP, such as in 1973 and 2010, were associated with particularly large uncertainty amongst the models. These





large differences on inter-annual timescales led to differences in cumulative NBP over 1901–2018, varying between -4.7 PgC
(OCN) and 9.5 PgC (LPX-Bern; see fig. 1b). Two of the DGVMs simulated a net carbon source over the Australian continent
from 1901–2018 while the remaining models simulated a net sink ranging from negligible through to 9.5 PgC.

Across the 20th Century, the TRENDY models simulated very different amounts of $C_{Veg}$. For example, JSBACH simulated
$\sim$2.5 PgC while JULES-ES and ORCHIDEE-CNP simulated more than $\sim$16 PgC (see fig. 1c). We used an estimate derived
from vegetation optical depth (VOD; see fig. 1c) to put the model simulations into context. Nine of the thirteen models predicted
a higher amount of $C_{Veg}$ than aboveground biomass estimated by the satellite. Uncertainty in the VOD data was not available,
but if we conservatively assume $\pm$ 30%, at least 7 of the 12 models were outside this range. However, we note that higher
$C_{Veg}$ in comparison to aboveground biomass might also result from the fact that $C_{Veg}$ includes above- and belowground
biomass. The models also showed very different long term behaviour. For example, some DGVMs simulated a steady state
$C_{Veg}$ (e.g. JSBACH and LPX-Bern) through 1901–2018, while other models displayed a strong decrease in $C_{Veg}$ until 1970 and
subsequently increased slightly (e.g. OCN and ISBA-CTRIP). VISIT simulated a sudden jump in $C_{Veg}$ around 1975. Similarly,
$C_{Soil}$ varied strongly between the models from 11.9 PgC (JSBACH) up to 64.4 PgC (JULES-ES). Most of the models did not
show a change over time while three models increased in $C_{Soil}$ (CABLE-POP, ISAM and LPX-Bern). In short, figure 1 shows
that the 13 TRENDY models simulated different inter-annual variability, contrasting cumulative NBP and inconsistent short
and long term $C_{Veg}$ and $C_{Soil}$.

**3.2 Response to atmospheric $CO_2$ concentration, climate and land-use change**

We used three sets of TRENDY simulations (see appendix figure B1) to separate the effect of three drivers — atmospheric
$CO_2$ concentration, climate and land-use change — on the Australian carbon cycle. Figure 2 shows cumulative NBP over
1901–2018 for the three simulations $CO_2$, $CO_2$ + CLIM and $CO_2$ + CLIM + LUC. Summed over the years 1901–2018, the
TRENDY models showed high variability in the response to the different external drivers. The models agreed on the sign of the
response to atmospheric $CO_2$ and the combined effect of atmospheric $CO_2$ and varying climate, such that by 2018, all models
accumulated NBP, both on regional and continental scales. However, the response of cumulative NBP to increasing atmospheric
$CO_2$ differed strongly among the models with an increase in NBP ranging from 2.5 PgC up to 10 PgC over all of Australia.
Varying climate in combination with increasing atmospheric $CO_2$ concentration led to stronger increases in cumulative NBP
for some models (e.g. LPX-Bern and VISIT, see fig. 2; all regions except tropics and mediterranean), or decreases (e.g. ISAM
and ORCHIDEE-CNP). Lastly, land-use change led to the the greatest variation among the TRENDY-models with differences
both in sign and magnitude (–3.3–8.5 PgC). Eight out 13 models simulated a decrease in cumulative NBP compared to the $CO_2$
+ CLIM run for all regions. Imposed land-use turned positive cumulative NBP in the $CO_2$ + CLIM simulations into negative
cumulative NBP in the $CO_2$ + CLIM + LUC simulation for OCN and ORCHIDEE-CNP. Only ISAM accumulated more NBP
in the $CO_2$ + CLIM + LUC simulation compared to the $CO_2$ + CLIM run in most regions.



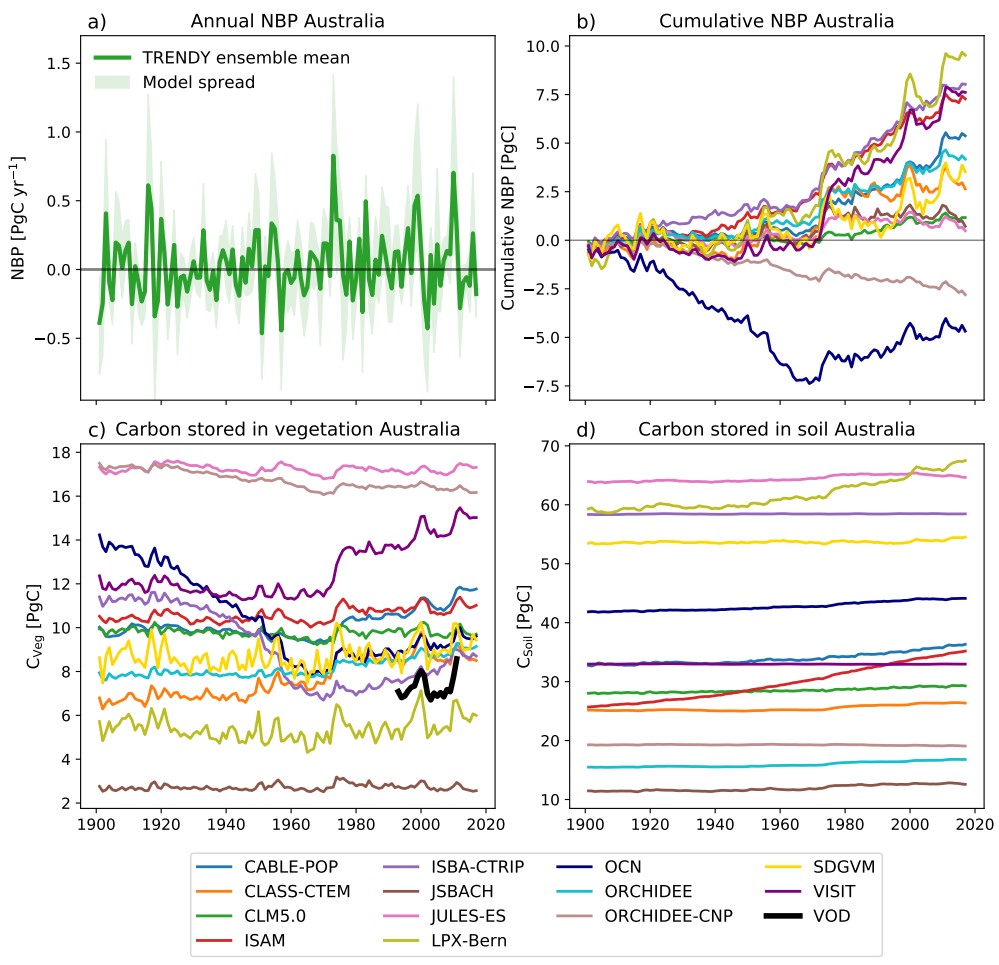

**Figure 1.** Net biome production (NBP), carbon stored in vegetation ($C_{Veg}$) and carbon stored in soil ($C_{Soil}$) for the TRENDY model ensemble for the $CO_2$ + CLIM + LUC run. Positive values in NBP represent a net carbon sink, negative values are a net carbon source. The solid green line in a) shows the ensemble mean of total annual NBP and the shaded green area shows the range across models. b) shows the cumulative net biome production for each model summed over Australia. c) and d) show $C_{Veg}$ and $C_{Soil}$ respectively summed over Australia for each model. The black line in c) shows the a satellite derived estimate ('VOD'; Liu et al., 2015). Note that we define the first month of the year as July and the last month of the year as June for NBP and follow calendar years for the carbon pools.

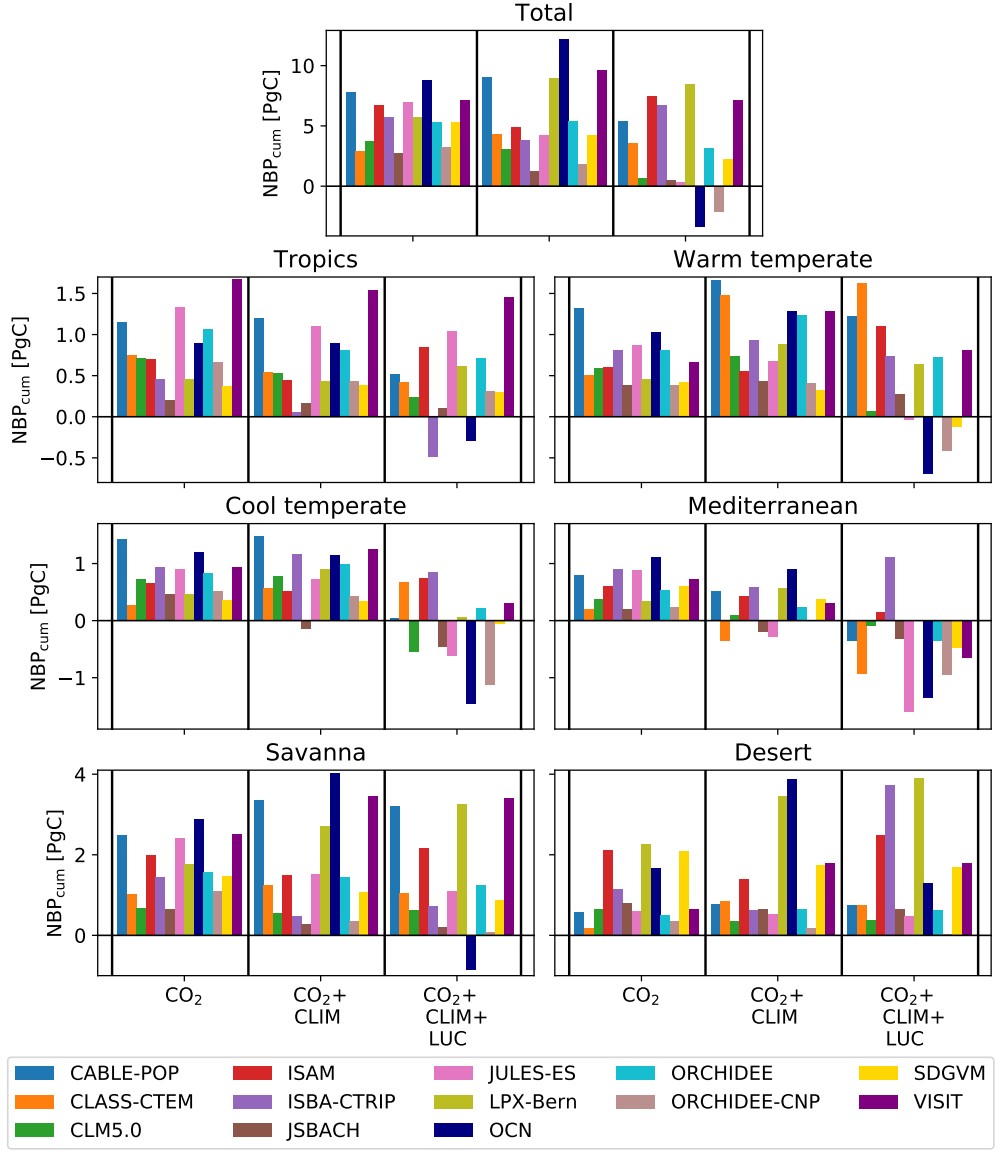

**Figure 2.** Cumulative net biome production (NBP) from 1901–2018. The first group in each panel shows the $CO_2$ effect on accumulated NBP ('$CO_2$'; i.e. transient $CO_2$ forcing, time-invariant climate and land-use-mask). The second group ('$CO_2$ + CLIM') shows the combined effect of a transient $CO_2$ and climate forcing, i.e. transient $CO_2$ and climate forcing, time-invariant land-use-mask). The third group ('$CO_2$ + CLIM + LUC') shows the combined effect of a transient $CO_2$, climate forcing and land-use change, (i.e. transient $CO_2$ and climate forcing and land-use-mask). Regions are defined according to (see appendix figure B4; Haverd et al., 2012). We define the first month of the year as July and the last month of the year is defined as June. Note that the y-axis limits differ between the panels.



## 3.3 Response to modes of climate variability

The climate in Australia, especially the precipitation patterns, is strongly influenced by different modes of variability such as ENSO and the IOD. Therefore, the modes of variability can be expected to influence the IAV of the terrestrial carbon cycle over Australia. To further understand the response to the climate forcing, we decomposed the model response to the different modes of variability (see appendix table B2), specifically to positive and negative IOD events (see fig. 3 'pIOD' and 'nIOD'), and El Niño and La Niña events (see fig. 3 'El Niño' and 'La Niña'). Negative and positive NBP values that are neither associated with ENSO nor with the IOD (see fig. 3 'other negative' and 'other positive') as well as the response of the models to the ten driest and wettest years ('Driest years' and 'Wettest years') are also shown in figure 3.

In general, the individual models agreed on the sign of NBP for the different climate modes of variability with negative NBP for positive IOD and El Niño events, and positive NBP for negative IOD and La Niña events. The two 'other' groups (negative and positive) covered a similar value range compared to the modes of variability. For both IOD- and ENSO-events as well as for the 'other negative/positive' categories, the median values across years were similar among the models. Interestingly, the model responses in the driest and wettest years led to the largest range among the models in both median and IAV compared to the other six panels. This highlights the importance of looking beyond climate modes of variability to understand responses of the Australian terrestrial carbon cycle.





**Figure 3.** Net biome production (NBP) during positive IOD ('pIOD'), negative IOD ('nIOD'), El Niño and La Niña events (see appendix table B2). All negative NBP values that are neither associated with pIOD nor El Niño are classified as 'other negative'; all positive NBP values that are neither associated with nIOD nor La Niña events are classified as 'other positive'. 'Driest years' show NBP values associated with the ten driest years, and 'Wettest years' show NBP values associated with the ten wettest years (based on precipitation anomalies averaged over Australia). We define the first month of the year as July and the last month of the year is defined as June. Note that the y-axis limits differ between the panels.





## 3.4 Seasonal productivity and phenology

To examine differences in simulated carbon uptake, we evaluated the simulated seasonal cycle. Figure 4 shows that the TRENDY models varied in the timing and the magnitude of peak productivity for the different vegetation regions. We compared the TRENDY models to the GPP estimate derived from solar-induced chlorophyll fluorescence ('GOSIF-GPP'; Li and Xiao, 2019). We note that because direct measurements of GPP at continental scales do not exist (i.e. GOSIF-GPP is not directly observed GPP), we constrained our comparison to an indicative evaluation of differences. Most models simulated higher productivity for the tropics, savanna and warm temperate regions compared to GOSIF-GPP. For all regions except the desert, most models were not within the uncertainty band of the GOSIF-GPP satellite estimate (grey band). The timing of peak productivity varied among the TRENDY models and GOSIF-GPP: most models matched peak productivity for the tropics, Savanna and desert. In contrast, for the cool temperate and Mediterranean regions, most models lagged peak productivity estimated by GOSIF-GPP by one to three months. As the SIF–GPP relationship was derived using a fixed relationship between eddy covariance and satellite SIF, it is possible that the model data mismatch implies a missing species sensitivity to water stress in the SIF data. However, the consistent timing in peak SIF and satellite LAI (see appendix figure B2) tends to imply that the lags in phenology relate to differences in assumed/emergent vegetation cover (see below).



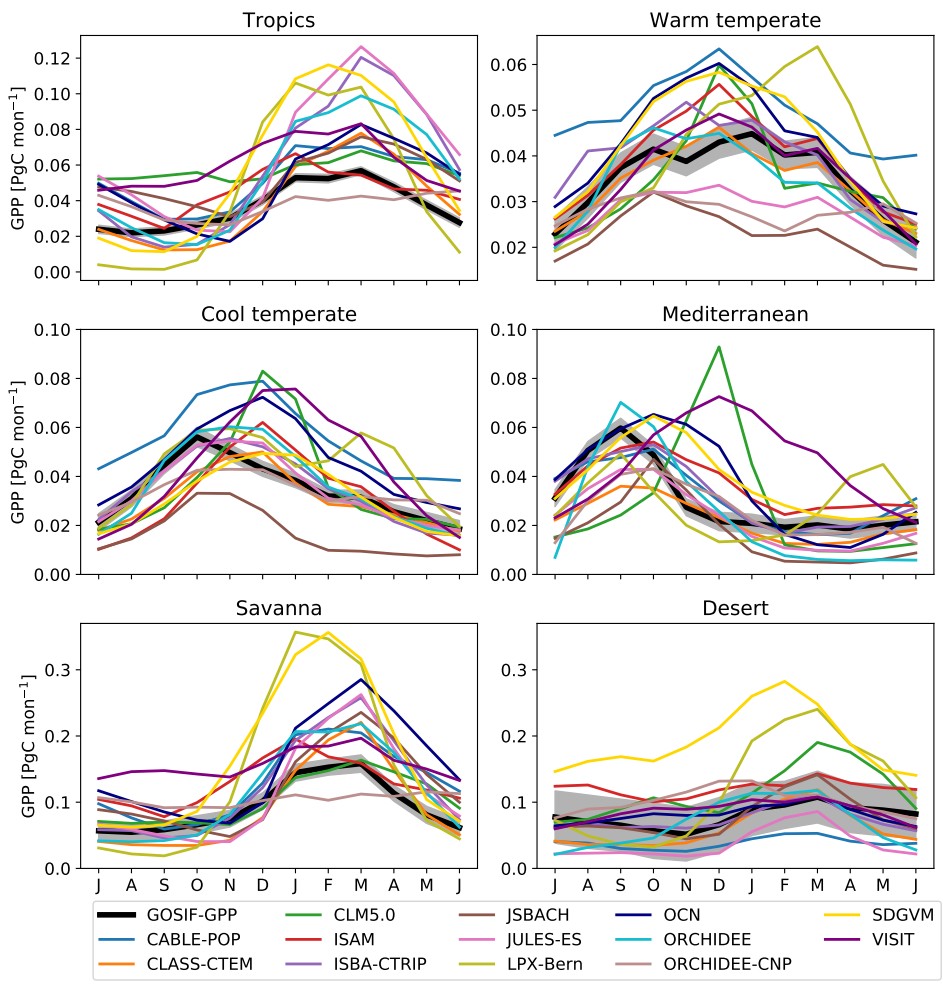

**Figure 4.** Seasonal cycle for satellite derived gross primary production (GOSIF-GPP; black) and the associated uncertainty (i.e. ± 1 standard deviation) as well as the individual TRENDY models averaged over 2000–2017. Regions are defined according to appendix figure B4. Note that the y-axis limits differ between the panels.

### 3.5 Apparent carbon residence time

Relative to the average over 1901–1930, all models simulated a similar rate in increase in NPP (see fig. 5a). The apparent carbon residence time ($\tau$), i.e. the balance between growth of plant tissues (leaves, wood, roots) and the turnover of these tissues, varied between 2 and 14 years averaged over Australia. Models that simulated high $C_{Veg}$ (compare fig. 1) also had high values for $\tau$ (e.g. ORCHIDEE-CNP and JULES-ES) and vice-versa (e.g. JSBACH and LPX-Bern). Some models did not simulate changes in $\tau$ over time (e.g. CLM5.0, JSBACH, JULES-ES, LPX-Bern) while other models simulated decreases. In particular, the ISBA-CTRIP and OCN models showed a strong decline: from ~six years (1900) to four years (1960) before


leveling off (see fig. 5c). Even though NPP increased for ISBA-CTRIP and OCN, $C_{Veg}$ significantly declined for these two models as well due to the decrease in $\tau$. The remaining models either balanced increasing NPP and decreasing $\tau$ so that $C_{Veg}$ did not show a strong trend (e.g. CLM5.0, JSBACH, JULES-ES, LPX-Bern) or increased in $C_{Veg}$ because the effect of change in NPP was greater than the decline in $\tau$ (e.g. CLASS-CTEM, ORCHIDEE, VISIT).

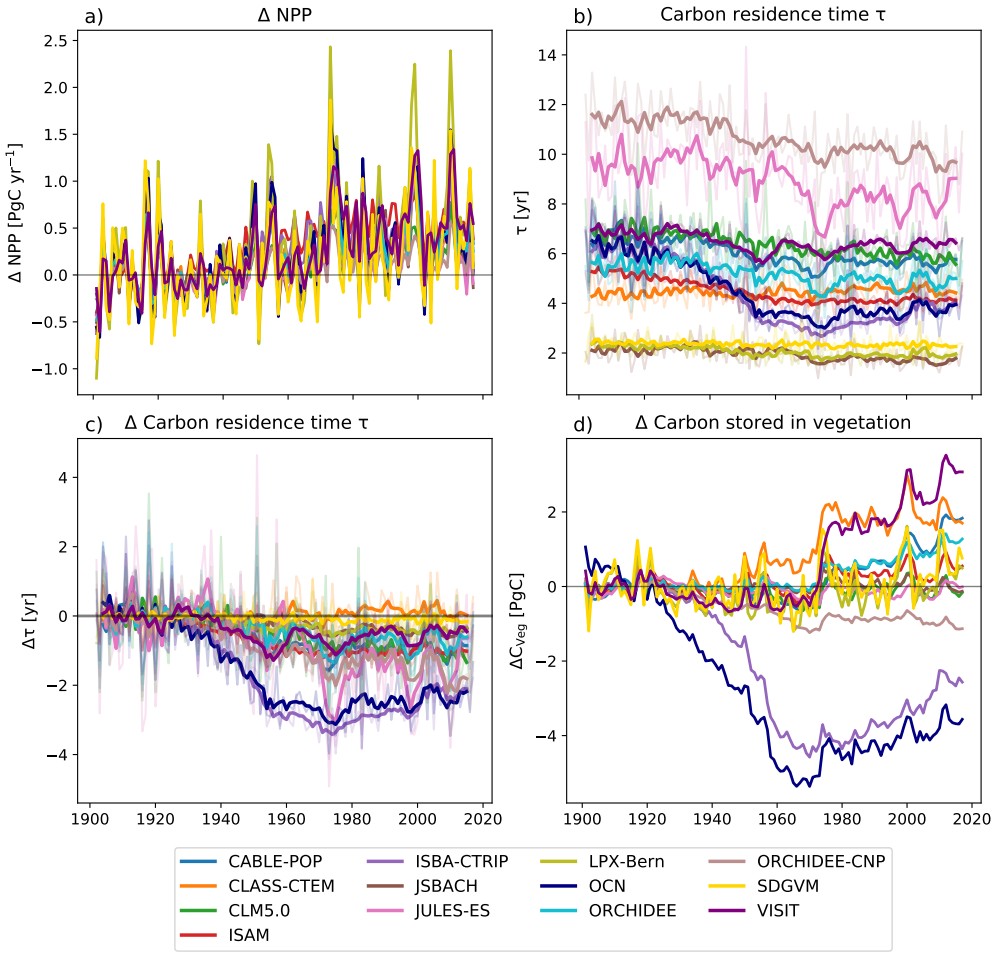

**Figure 5.** Change in net primary production (NPP) in comparison to the 1901–1930 average (a), total apparent carbon residence time in vegetation $\tau$ (b), change in $\tau$ in comparison to the 1901–1930 average (c) and change in carbon stored in vegetation ($C_{Veg}$) in comparison to the 1901–1930 average (d). The solid lines in panel b) and c) show the 5-year-moving average, the shaded lines are the original data.

**3.6 Northern Australian Tropical Transect (NATT)**

To better understand differences in modelled carbon fluxes, we examined simulations along a rainfall gradient: the Northern Australian Tropical Transect (NATT). Figure 6 shows the probability density functions for NEE, GPP and TER for the wet



(November–April) and dry (May–October) seasons across the NATT. During the dry season, as the sites become drier, the mean position of the GPP and TER distributions shifted closer to zero and the distribution spread narrows. By contrast, the shape of

the distributions varied strongly among the models for all variables: some models (e.g ISAM; LPX-Bern for dry season NEE and GPP) simulated high peak probability densities and narrow distributions while others (e.g. JULES-ES and OCN) displayed a flatter distribution. During the wet season, some models were more productive than observed for all sites (CABLE-POP, JSBACH, JULES-ES and SDGVM). Overall, there was no clear systematic pattern in the arrangement of the models across the NATT, even when models simulated similar distributions for NEE, they often did so for different reasons (i.e. contrasting

trade-offs in carbon uptake/respiration).




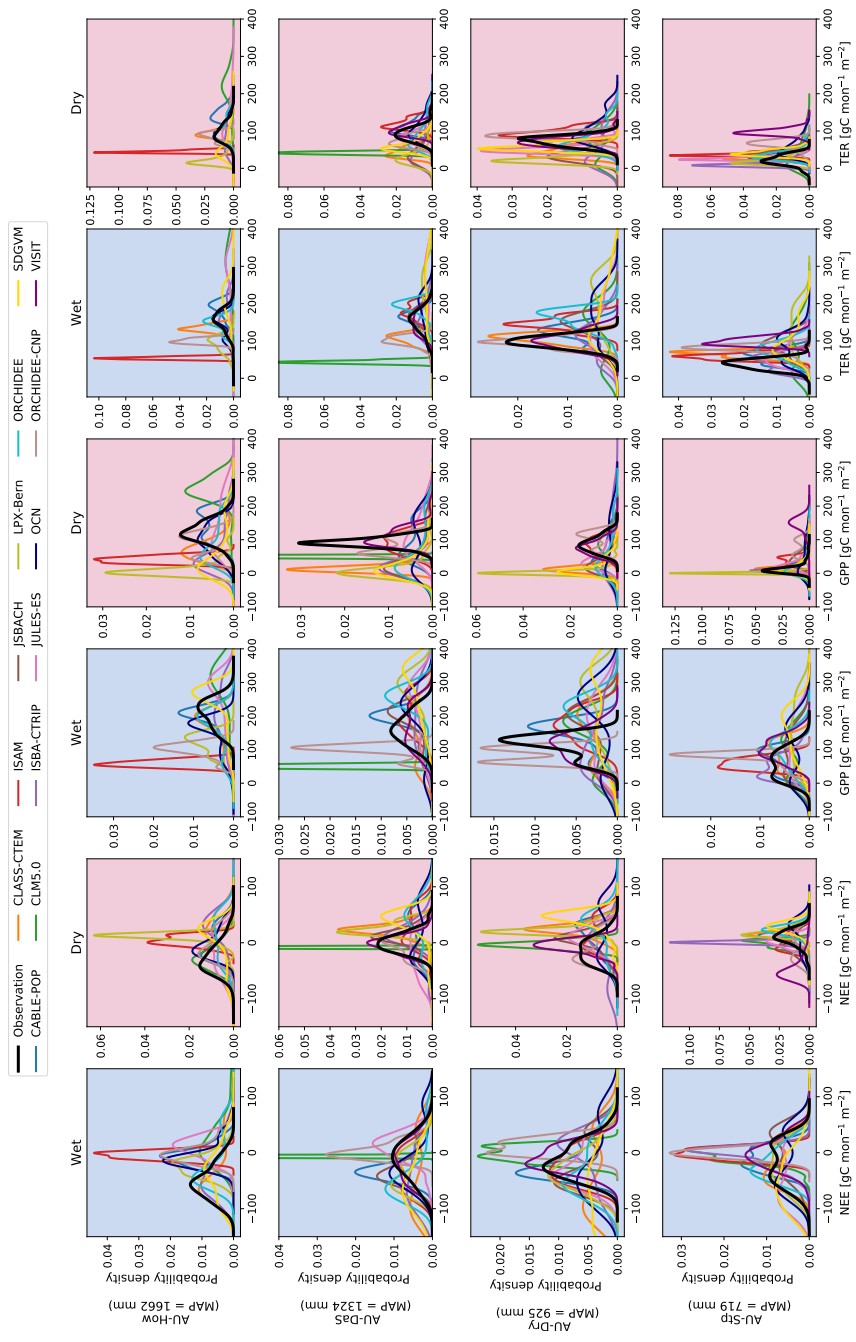

**Figure 6.** Probability density functions for observed and modelled net ecosystem exchange (NEE), gross primary production (GPP) and terrestrial ecosystem respiration (TER) for the for four fluxsites: Howard Springs (AU-How), Daly Uncleared (AU-DaS), Dry River (AU-Dry) and Sturt Plains (AU-Stp). These sites reflect a strong mean annual precipitation (MAP) gradient (provided for each row). We show the monthly distribution of each variable for the wet (November–April) and dry (May–October) seasons. The observed variables originate from eddy covariance data collected by the TERN-OzFlux facility. The probability density function is based on kernel density estimates using Gaussian kernels with bandwidth selection following Scott (1992).





## 3.7 Fire-related $CO_2$ fluxes

Fire is an important component of the carbon cycle in many of Australia's ecosystems. At the monthly timescale, the seven DGVMs that simulated fire dynamics, significantly underestimated the fire $CO_2$ emissions associated with extreme events compared to the GFED4s and CAMS-GFAS satellite products. However, the timing of simulated peak fire events was synchro-
nised with the observations for some models (see fig. 7a). Fire weather can be associated with ENSO-cycles in Australia (e.g. Harris and Lucas, 2019); figure 7a however did not indicate a clear connection between ENSO-cycles and extreme fire events. The first two peaks in observed fire emissions occur during El Niño events; the two highest peaks in fire $CO_2$ emission according to CMAS-GFAS instead coincided with the 2011 La Niña event and with the ENSO-neutral year 2012/ 2013. Figure 7b) shows that all models simulated lower variability compared to the satellite derived estimates with similarly wide but less peaked
probability density functions. The peak was shifted to the right of the observations for some models (LPX-Bern, SDGVM and VISIT; see fig. 7b). Aggregated to annual values, the TRENDY models generally underestimated the fire $CO_2$ emissions and did not capture the variability in, or timing of, extreme fire years (see fig. 7c). CLASS-CTEM, JSBACH and ISBA-CTRIP captured some features of the variability in the satellite derived observations on monthly timescales with significant moderate correlation coefficients (see appendix table B3) but it is hard to reconcile the other four models with either of the observational
products.

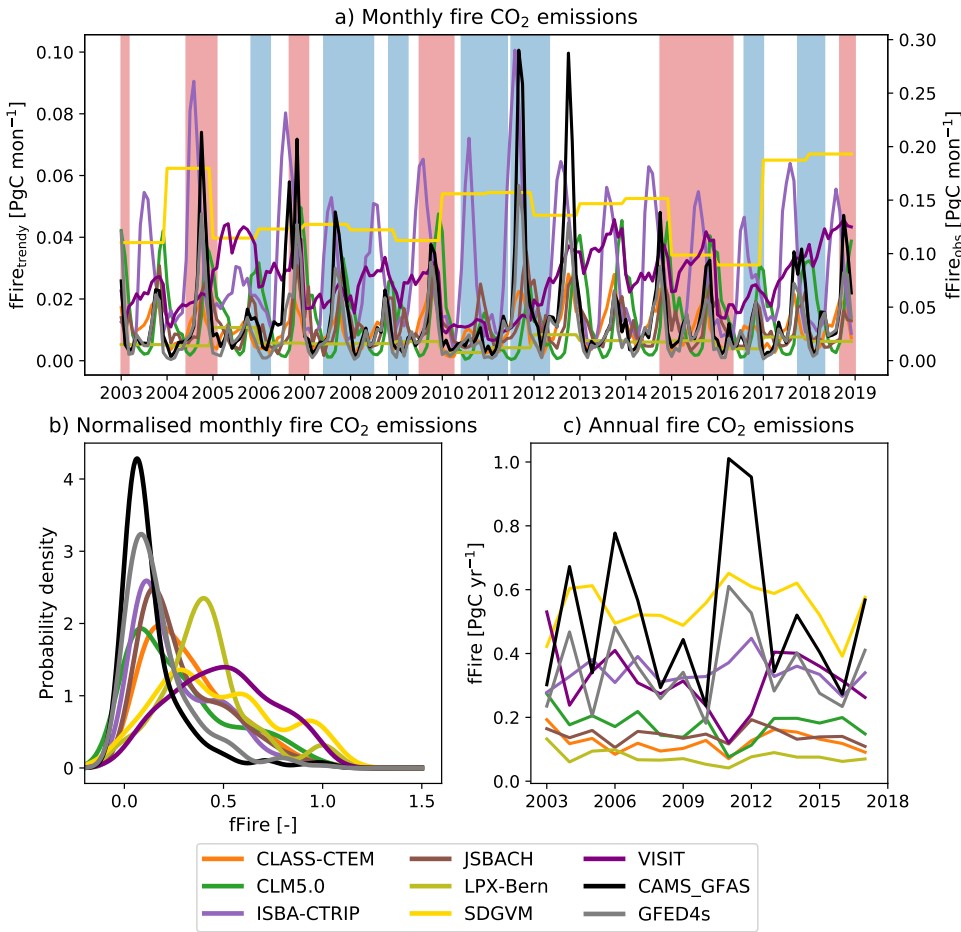

**Figure 7.** Monthly (a, b) and annual (c) fire $CO_2$ emissions of seven TRENDY models and two satellite derived estimates (CMAS-GFAS and GFED4s). Blue shaded areas show La Niña events, red shaded areas show El Niño events. a) and c) show the total fire $CO_2$ emissions, b) shows the probability density function based on kernel density estimates using Gaussian kernels with bandwidth selection following Scott (1992) of the normalised monthly fire $CO_2$ emissions. Note that the satellite derived estimates have a different y-axis for the monthly fire $CO_2$ emissions than for the TRENDY models.

## 3.8 Land cover fraction

Figure 8 shows the vegetation fraction, either prescribed or simulated dynamically, from the TRENDY models averaged over Australia. We grouped the model-specific PFTs into ten groups (see methods). In general, the average landcover varied quite strongly among the models. The average fraction of Australia covered by natural vegetation differed between 24.9% (LPX-Bern) up to 93.7% (VISIT). The vegetation composition also displayed large differences with some models simulating high tree fractions around 40% (ORCHIDEE and ORCHIDEE-CNP) while other models had low fractions around 2% (e.g. ISAM, and SDGVM). Models accounting for shrubs populated 3.5% (ISBA-CTRIP) up to 55% (VISIT) of Australia with shrubs.





Similarly, land populated by grasses covered a large range between 4% up to 50% (VISIT and OCN, respectively) and the fraction of C3 grasses either exceeded land covered by C4 grasses or vice versa. Most models showed a relatively small

proportion of agricultural landcover except for LPX-Bern. Lastly, models accounting for bare landcover fraction (i.e. bare ground or desert) defined 0.2–51% of Australia as bare (VISIT and SDGVM, respectively).

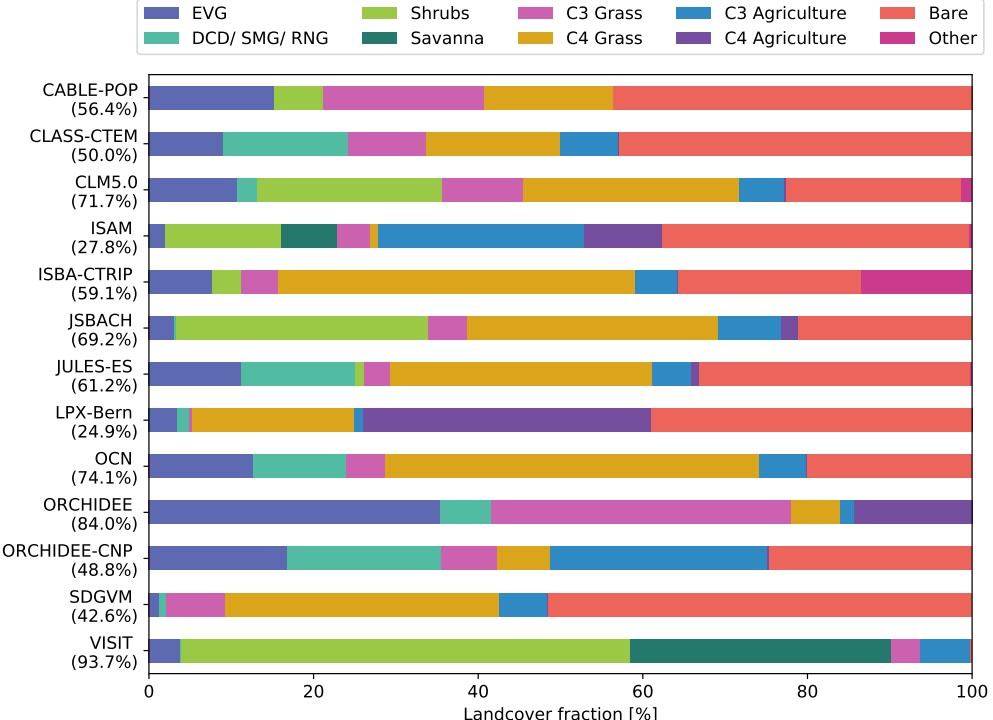

**Figure 8.** Landcover fraction in % averaged over 1989–2018 over Australia. EVG is the sum of all evergreen tree plant functional types (PFT)s, DCD/ SMG/ RNG is the sum of all deciduous, summergreen and raingreen tree PFTs, Shrubs is the sum of all shrub PFTs, Savanna represents the Savanna PFT, C3 Grass and C4 Grass include all natural C3 and C4 grasses, respectively, C3 and C4 agriculture represent all C3 and C4 crops and pasture PFTs, respectively, Bare is the sum of bare and desert landcover types and Other includes landcover types that are not captured by the above group, such as cities/ urban, and lakes. The numbers in brackets represents the land covered by natural vegetation (sum of EVG/ RNG, DCD/ SMG, Shrubs, Savanna and C3 and C4 grasses). Note that not all models account for shrub, savanna, and agriculture PFTs or for explicit bare and desert PFTs.

Figure 9 shows the spatial distribution of the fraction of all herbaceous vegetation, the predominant vegetation in Australia, averaged over 1989–2018 for the TRENDY models. The models displayed different patterns in land covered by grass and crops. For example, VISIT had a high grass fraction in almost all regions with values close to 1, whereas other models showed

dense grass cover fractions in the Tropics, and around the East and West of Australia (e.g. ISAM, ISBA-CTRIP, LPX-Bern, and SDGVM). Some models simulated a more even distribution of grass cover across Australia with either relatively high (JSBACH) or low fractions (CLASS-CTEM, JULES-ES, and ORCHIDEE-CNP). In contrast, the grass cover fraction for





ORCHIDEE generally increased moving from the coastal areas to the interior of the continent. The tree fraction also varied strongly across the models. Most models had high fractions along the East Coast and in the tropics and no significant tree
growth in other places, similar to the MODIS satellite derived tree cover fraction. In contrast, OCN had a low tree cover fractions along the coastline but this fraction increased towards the interior of Australia, with a higher distribution in the South West and South East (see appendix figure B5).

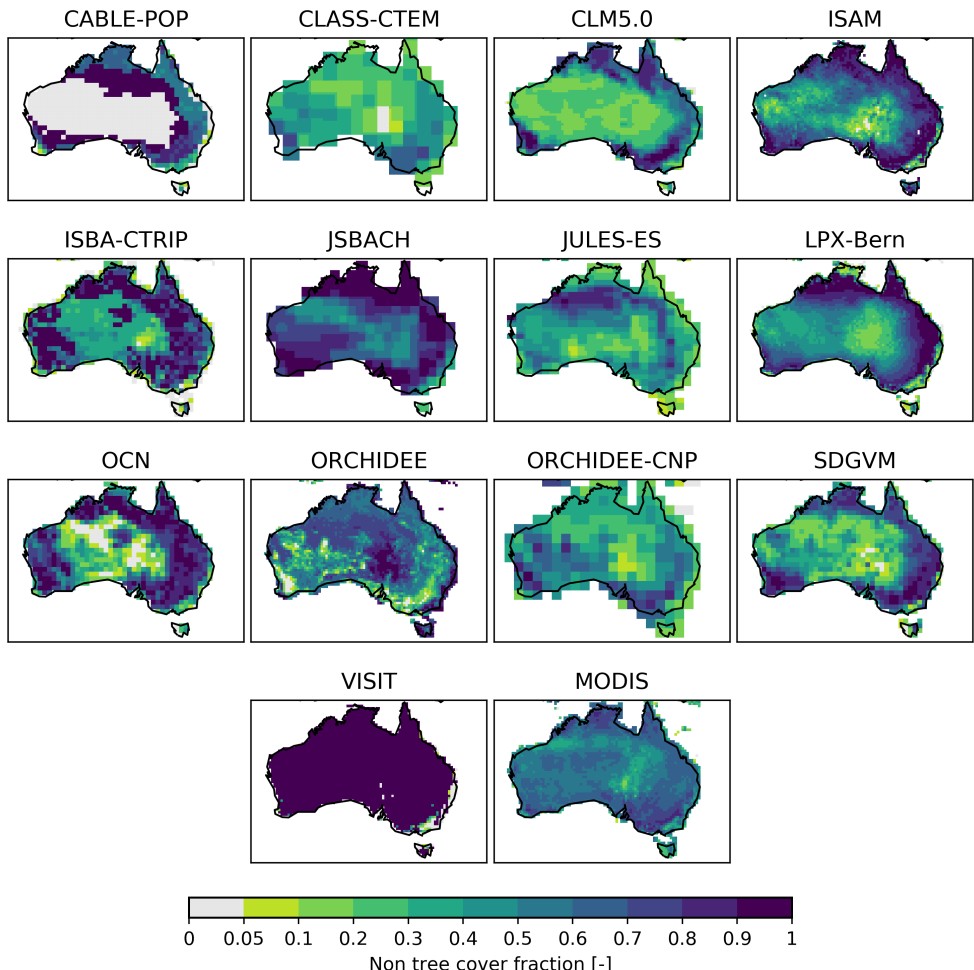

**Figure 9.** Landcover fraction of non tree vegetation averaged over 2001–2018. Non tree vegetation here shows the sum of all natural C3 and C4 grasses as well as C3 and C4 pasture and agriculture, and savanna PFTs in the TRENDY models. The MODIS panel shows the non tree fraction variable from the MODIS/Terra vegetation continuous fields dataset. White areas are missing values.





## 4 Discussion

On average, Australia's simulated contribution to the global NBP anomaly was 17%, with greater contributions when the global
sink (excluding Australia) was small, or in years with above average precipitation in Australia (e.g. in a La Niña years; Poulter
et al., 2014). Depending on the model, Australia's share of global cumulative NBP could be as high as ∼53% (SDGVM).
Given Australia's significant contribution to the global terrestrial carbon sink on inter-annual timescales, our goal was to assess
the skill of state-of-the-art DGVMs applied to Australia.

Despite being forced with identical meteorology, the 13 DGVMs from the TRENDY v8 ensemble simulated markedly
different representations of Australia's terrestrial carbon cycle. The maximum difference in annual NBP among the models
reached up to 1.3 PgC yr$^{-1}$ (see fig. 1a). Importantly, uncertainties associated with IAV in simulated NBP accumulated over
time, leading to large differences in cumulative NBP among the models by 2018 (see fig. 1b). The TRENDY models also
simulated very different amounts of carbon stocks in vegetation (fig. 1c). While the satellite estimate of $C_{Veg}$ lie within the
range simulated by the models, individual model estimates of $C_{Veg}$ varied widely (averaged over 1901–2018 from around 2.7
PgC for JSBACH to 17.2 PgC for JULES-ES). Similarly, averaged over 1901–2018, simulated $C_{Soil}$ differed strongly from
11.9 PgC (JSBACH) to 64.4 PgC (JULES-ES).

We identified the key reasons for model differences in NBP related to the processes land cover/land-use change, rising
atmospheric $CO_2$ concentration, carbon residence time in the vegetation, climate modes of variability and fire processes.

*Landcover*

We found that the models either prescribed or simulated very different fractions of woody and herbaceous cover (see fig. 9
and appendix figure B5), and that these discrepancies likely explained a large proportion of the model divergence in simulated
$C_{Veg}$ and $\tau$ (see below). Notably, the two models with the lowest $C_{Veg}$ had a comparably low tree cover (JSBACH and
LPX-Bern; see fig. 8) and a high proportion of bare land. In contrast, the two models with the highest $C_{Veg}$ (JULES-ES
and ORCHIDEE-CNP) simulated a high tree cover fraction. However, figure 8 also implies that seemingly similar PFTs were
parametrised in different ways in the individual models. For example, CLM5.0 and JSBACH simulated a similar fraction of land
covered by natural vegetation with a comparable vegetation composition comprising trees, shrubs, natural grass, agriculture
and bare land. However, CLM5.0 stored around four times more carbon in vegetation compared to JSBACH.

Land-use and land-cover change is one of the major drivers of the terrestrial carbon sink on global and regional scales
(e.g. Pugh et al., 2019; Huntzinger et al., 2017). Over Australia, the largest divergence in cumulative NBP was associated
with the implementation of land-use change (LUC; see fig. 2). Overall, the TRENDY models simulated different magnitudes
of cumulative NBP in response to LUC, ranging from -15.5 PgC (OCN) up to 2.9 PgC (ISBA-CTRIP). For two models,
accounting for LUC turned almost all regions from a carbon sink to a carbon source (OCN, ORCHIDEE-CNP).

Whilst DGVMs were developed to understand the interactions between natural vegetation and the atmosphere, the insight
that land-use is a major driver of the terrestrial carbon sink has driven efforts to incorporate land use change into DGVMs
(Marquer et al., 2018; Pongratz et al., 2018). Differences in the models' response to LUC results from process inclusions,
parametrisations and the model-specific interpretation of the land-use forcing (e.g. change in vegetation composition, as well





as wood/crop harvest). Although LUC information was taken from the Land-Use Harmonization 2 (LUH2) guidelines, the modelers had to decide how to implement it. For example, the LUH2 suggests that all natural vegetation should be cleared for the conversion to managed pasture while the conversion to rangeland only requires the transformation of forest to (managed) grassland. Other natural vegetation, such as grass- or shrubland, were used for animal browsing without any transformation of the land cover type (Ma et al., 2020). Furthermore, the fraction of croplands reconstructed by LUH2 increased markedly in South West Australia and South East Australia. This increase in crop fraction was associated with increased harvests and consequently expected to reduce NBP. Some models exhibited this feature to varying degrees (JULES-ES, OCN, and ORCHIDEE-CNP) while other models barely increased the crop fraction and impact on NBP (CABLE-POP, ORCHIDEE and VISIT; see appendix figure B6). These different representations of LUC led to emergent differences (e.g. in physiology as the models transition between PFTs and agricultural management) when harvesting was imposed on models simulations.

*Atmospheric $CO_2$ concentration*

All the models simulated an increase in cumulative NBP in response to increasing $CO_2$ but with very different magnitudes (+2.46–10.13 PgC from 1901–2018). This was in line with studies that have identified increasing $CO_2$ as the main driver of change in the global terrestrial carbon sink and find that model disagreement was largest due to differing sensitivities to $CO_2$ in cumulative NBP (e.g. Huntzinger et al., 2017; Arora et al., 2020; Walker et al., 2020). The incorporation of nutrient cycles in models and DGVMs in particular, tends to result in a reduced sensitivity to $CO_2$ (e.g. Smith et al., 2014; Zaehle, 2013; Thomas et al., 2013; Meiyappan et al., 2015; Meyerholt et al., 2020). In a global analysis, Huntzinger et al. (2017) found that models incorporating an nitrogen cycle tended to have a lower net carbon sink than models without nitrogen constraints. Similarly, two model based analyses in the context of $CO_2$ manipulation experiments, found lower simulated net primary productivity responses in models that incorporated nitrogen and phosphorus cycles (Medlyn et al., 2016; Fleischer et al., 2019). Interestingly, this finding does not hold for our study: the three models with the highest cumulative NBP all included a nitrogen cycle. A possible explanation may be that Australian soils are considered to be phosphorus, rather than nitrogen limited (Du et al., 2020; Lambers et al., 2015) due to weathering processes and high phosphorus sorption capacities (Beadle, 1966; Wild, 1958). Results from an ecosystem-scale $CO_2$ manipulation experiment (Ellsworth et al., 2017) showed that phosphorus availability limited biomass growth in an Australian woodland, although leaf-level photosynthesis was observed to have consistently increased (Jiang et al., 2020; Ellsworth et al., 2017; Yang et al., 2020). Despite the importance of phosphorus availability in Australia, only two of the TRENDY ensemble incorporated an interactive phosphorus cycle and these models (CABLE-POP and ORCHIDEE-CNP) did not show consistent behaviour. It is likely that improved modelling of the phosphorus cycle and in particular, the relative 'openness' of this cycle and the flexibility of the plant tissue stoichiometry, will remain key to accurately simulating the time evolution of Australia's carbon cycle (Medlyn et al., 2016).

*Apparent carbon residence time in vegetation*

The capacity of the terrestrial vegetation to store carbon depends on the magnitude of the input carbon flux and the carbon residence time in plant tissues ($\tau$; Luo et al., 2003). All models simulated increases in NPP at similar rates compared to their initial NPP (1901–1930 average; see fig. 5). As the other models simulated a similar change in the rate of carbon uptake throughout the period, divergence in the change of $C_{Veg}$ instead resulted from differences in the change of $\tau$. $\tau$ depends on





the turnover of plant tissues, the simulated mortality rates and mortality induced by competition. Each of these processes can be affected indirectly through shifts in vegetation composition (Friend et al., 2014). Previous studies have found that $\tau$ was a key source of model uncertainty shared amongst DGVMs (Friend et al., 2014; Pugh et al., 2020). In an analysis of CMIP5

models, Carvalhais et al. (2014) found that less than a quarter of the models were within the range of of observed $\tau$ in tropical Australia and parts of the Northeastern Savannas. Similarly, we found that $\tau$ in the TRENDY models was associated with large uncertainty: between 2 and up to 14 years over Australia. Models that simulated high vegetation carbon storage also had comparably long $\tau$ and vice versa. These differences in modelled $\tau$ imply that the TRENDY models assume different parametrisations associated with their PFTs, mortality and/ or simulated different vegetation compositions and this is an area

requiring future evaluation.

*Climate and modes of climate variability*

Variability in Australia's climate can influence the terrestrial carbon sink directly (e.g. fire, droughts or increased water availability; Keenan and Williams, 2018; Ma et al., 2016) or indirectly via interactions with nutrient availability or fire. For example, models that include interactive fire modules might directly (e.g. wind speed driving fire spread; temperature threshold

to allow for fire) and indirectly (fuel availability and moisture) depend on climate variables, and altered fire patterns could further affect the simulated terrestrial carbon sink. Overall, we found that the effect of climate on inter-annual NBP was relatively small, with the exception of the Mediterranean region (see fig. 2). This is in general agreement with Huntzinger et al. (2017) who found that long term trends in climate were too small to significantly alter the simulated terrestrial carbon sink on long timescales and was less important factor than $CO_2$ fertilisation, nitrogen deposition (depending on whether the models

include an interactive nitrogen cycle) and land-use change. While climate variability might not influence Australian carbon cycle trends on long timescales, inter-annual weather variability, especially the amount and timing of precipitation, is a strong control on inter-annual variability in NBP.

Australia's weather is influenced by different modes of variability with El Niño and pIOD events tending to result in below average precipitation in south eastern Australia (e.g. Ummenhofer et al., 2011). The reduced water availability can be expected

to reduce NBP or even turn Australia from a carbon sink to a carbon source (e.g. Ma et al., 2016). In contrast, nIOD and La Niña events are associated with above average rainfall and consequently, an enhanced carbon sink (e.g. Ma et al., 2016; Bastos et al., 2013). The TRENDY models mostly agreed on sign of the NBP anomaly and simulated negative median values for both pIOD and El Niño events and positive median values for nIOD and La Niña events. The sign of the NBP flux was not unambiguously driven by either ENSO or IOD but was also influenced by other variability (see fig. 3 'other negative' and 'other

positive'). The interquartile range for the category 'other negative' was similar to those for the pIOD and El Niño categories and displayed more negative outliers than the other modes of variability. This implies that years with extremely low NBP were not necessarily associated with pIOD or El Niño but instead driven by other modes of climate variability and/or periods that did not reach the threshold to be defined as an ENSO or IOD event. Especially for the nIOD, La Niña and other positive categories, the models varied strongly in the interquartile range, indicating the individual models simulated different responses

to the modes of variability. Studies showed that other modes of variability, such as the Southern Annual Mode (SAM), can influence the Australian weather patterns and consequently the terrestrial vegetation (e.g. Cleverly et al., 2016). We did not





include an analysis of SAM because these events are short lived (one–two weeks), but future work may examine the resulting impacts. We note that selecting years based on ENSO- and IOD-events does not completely isolate the effect of ENSO- or IOD-events on the terrestrial carbon cycle as these modes of variability are not independent and could have legacy impacts that last more than one year. Further, years where both an IOD and an ENSO event occur were accounted for in both the IOD and ENSO categories and therefore, double counted. An additional limitation of our approach was that we compiled the occurrence of ENSO and IOD events from different sources based on the sea surface temperature. Observations of sea surface temperature prior to 1960 however are associated with large uncertainty given they were mostly based on ship data (e.g. Deser et al., 2010). In addition, the Southern Hemisphere tends to be less well observed and spurious trends in reanalyses do occur (e.g. Hines et al., 2000), leading to an overall increased uncertainty in the sampling of data based on modes of variability.

Overall, dry and wet extremes in precipitation led to the largest differences among the models (see fig. 3 'Driest years' and 'Wettest years'), suggesting that the models' sensitivities to wet and dry extremes vary strongly. About half of the TRENDY ensemble had a stronger response to wet than to dry extremes. This is in line with studies that have noted an increased carbon uptake resulting from the asymmetry in the inter-annual distribution of rainfall as well as the asymmetry in the response of GPP to precipitation (e.g. Haverd et al., 2017). The ecosystem in Australia largely consists of vegetation that is adapted to drought (e.g. Cleverly et al., 2016; Li et al., 2018; De Kauwe et al., 2020). About half of the TRENDY ensemble displayed greater divergence during dry extremes, implying that models did not accurately capture process responses as water becomes limiting. This result is consistent with an a model intercomparison at an Australian woodland site, which showed disagreement was greatest in low-rainfall years (Medlyn et al., 2016). The TRENDY models not only simulated different carbon uptake sensitivities to precipitation but also simulated different seasonal phenology (see fig. 4), suggesting that differing model sensitivities to rainfall may result as much from the underlying simulated vegetation as the different mechanisms at play in wet/dry extremes. We further looked at the TRENDY output in context of the NATT, defined by a strong rainfall gradient ($\sim$ 720 to 1650 mm yr$^{-1}$). We found that for dry season GPP, the mean position of the distributions shifts closer to zero and the distribution narrows as the sites become more arid, implying that vegetation is only productive following rain (Whitley et al., 2016). By contrast, there was no clear systematic pattern in the responsiveness of simulated productivity during the wet and dry seasons.

*Fire-related CO$_2$ fluxes*

The seven TRENDY models that included fire outputs consistently underestimated satellite-derived peak monthly emissions and did not accurately capture the timing of peak fire CO$_2$ emissions. Importantly, we found that DGVMs did not accurately capture fire dynamics in Australia, even when aggregated annually (which should reduce timing biases). While land-use change has been identified to be a significant driver of global burned area (Teckentrup et al., 2019), we found that the effect of land-use change on fire CO$_2$ emissions in Australia was small (see appendix figure B7). Models either increase or decrease in fire CO$_2$ emissions due to land-use change but these changes were mostly smaller than 1%. All models simulated fire with varying degrees of complexity and derive fire CO$_2$ emissions based on model-specific burned area using emission factors based on Andreae and Merlet (2001) and Akagi et al. (2011). Consequently, model disagreements must originate from differences in simulated burned area.





Biases in burned area have been found to be associated with underlying vegetation composition (which determines fuel load; Teckentrup et al., 2019) and how vegetation cover changes with rising $CO_2$ and land-use change (through the conversion of natural vegetation to agriculture; Teckentrup et al., 2019). Model evaluations suggest that fire models broadly capture first-order patterns of emissions under present-day conditions (e.g. Li et al., 2019; Hantson et al., 2020), but simulations of seasonality

and inter-annual variability are associated with high uncertainty due to differences in process representation. We did not find evidence that models linked extreme fire events with ENSO as suggested by Harris and Lucas (2019). However, the limited availability of 'observed' fire $CO_2$ emissions makes it difficult to draw firm conclusions. In addition, while Harris and Lucas (2019) found that El Niño like conditions can be associated with extreme fire weather, they also emphasized that this depends on the regions and is most pronounced in eastern Australia whereas our analysis focused on all of Australia. Lastly, global

fire emission estimates based upon satellite observations are still associated with substantial uncertainty, as reflected by the considerable differences in spatial and temporal patterns between different data products (e.g. Li et al., 2019). To account for satellite-derived uncertainty, we included two satellite products, the GFED4s (van der Werf et al., 2017) and the CAMS-GFAS (Kaiser et al., 2012) datasets, that are derived based on different variables (burned area until 2016, subsequently active fire detections for GFED4s; fire radiate power for CAMS-GFAS). However, since both products rely on the MODIS sensor, they

are not independent and consequently only capture the uncertainty associated with satellite derived fire $CO_2$ emission estimates to a limited extent.

## 5  Future directions

Our analysis highlights considerable work is still required to improve our capacity to simulate the dynamics of Australian vegetation and the resulting terrestrial carbon cycle. This finding is consistent with previous model evaluations of NPP (Roxburgh

et al., 2004), carbon flux responses to rainfall (Whitley et al., 2016, 2017) and responses to elevated $CO_2$ (Medlyn et al., 2016).

When DGVMs are applied to Australia, the assumed plant traits are not adjusted to reflect differences in observed physiology (e.g. Bloomfield et al., 2019; Cernusak et al., 2011), investment in plant tissues (e.g. Togashi et al., 2015), how moisture limits plant function (e.g. De Kauwe et al., 2020; Li et al., 2018) or responses of photosynthesis and growth to temperature (Drake et al., 2015). For example, models typically assume much lower values for the key model parameter $V_{cmax}$ (the maximum

rate of carboxylation by the enzyme Rubisco; Rogers, 2013) than is commonly observed in Australia (e.g. Bloomfield et al., 2019; Cernusak et al., 2011), implying that DGVMs miss key carbon and water trade-offs. Whitley et al. (2017) previously stressed the importance of improving process representation of phenology, root-water uptake, rooting depth and fire to improve the representation of Savanna ecosystem dynamics in DGVMs. These Savanna ecosystem cover around 20% of the Australian landscape, but as this study implies (see fig. 6), progress has been limited, although recent model developments incorporating

competing rooting strategies and dynamic root growth (Sakschewski et al., 2020) may help this process. Australia has a high fraction of endemic vegetation (>90%; Chapman, 2006), structurally distinct vegetation including open canopy forests and woodlands (predominately Eucalyptus) and hummock grasslands and a dominance of Sclerophyll leaves (Box, 1996). Thus, there remains a significant opportunity in future work to directly test the current hypothesis implicitly embedded in DGVMs,



i.e. that Australia's vegetation is not distinct from other continents. The forthcoming AusTraits (https://ardc.edu.au/project/
austraits-a-curated-plant-trait-database-for-the-australian-flora/) database presents one possibility to examine how the use of
Australian specific plant traits may change DGVM simulations.

One core area in which model development should be focused relates to the processes that govern the vegetation cover.
In these models vegetation cover is either dynamically simulated (CABLE-POP, JULES-ES, LPX-Bern and SDGVM) or
prescribed (CLASS-CTEM, CLM5.0, ISAM, ISBA-CTRIP, JSBACH, OCN, ORCHIDEE, ORCHIDEE-CNP and VISIT).
Our study has highlighted major differences across models under historic forcing (see fig. 8 and 9), implying that future
simulations need to be interpreted cautiously. Haxeltine et al. (1996) demonstrated a realistic simulation of Australia's potential
vegetation cover with the BIOME2 model and yet, 25 years later, it is not clear that DGVMs have made significant progress.
One explanation for this apparent discrepancy may simply relate to focus of the modellers. In the Haxeltine et al. (1996) study,
the model explicitly aimed to reproduce Australia's potential vegetation cover, whereas the current TRENDY models are
instead focused on global applications. Nevertheless, our results imply that a one-size fits all approach does not appear to work
for Australia. Australia's National Vegetation Information System (https://www.environment.gov.au/land/native-vegetation/
national-vegetation-information-system) generates maps of both vegetation distribution as well as structure, which could be
used to evaluate and develop DGVM predictions of potential vegetation. However, a direct comparison to observed vegetation
distributions still requires model-specific rules to transform simulated foliar projected cover and height to vegetation classes.
Our results also showed that imposing land-use change adds further uncertainty to the vegetation distribution. Despite the
importance of improving simulated vegetation type, as Fisher et al. (2015) highlighted, it is not necessarily a straight forward
task but over-reliance on bioclimatic boundaries is not preferable moving forwards. Emergent differences in cover type may
also originate from parameter assumptions, which determine carbon uptake/loss, additional hypotheses that govern demography
and responses to fire. Haverd et al. (2016a) demonstrated promise in accurately predicting tree-grass partitioning in Australian
savannas using optimality theory to determine daily carbon allocation. However, this approach has yet to be incorporated into
any of the Australian land models (e.g. CABLE-POP), so a fuller evaluation is still required, including the resulting interactions
with a nutrient cycle. Kelley et al. (2014) demonstrated significant progress in both the simulation of fires in Australia and
the resulting grass-tree cover fractions, by incorporating PFT-specific fire (adaptive bark thickness) resistance and post-fire
epicormic resprouting into a DGVM. However, these features have yet to be incorporated into any current DGVMs when
applied to Australia.

In conclusion, there is a major opportunity in evaluating DGVMs using Australia as a laboratory. The strong inter-annual
variability in precipitation offers an opportunity to evaluate the simulated response of the Australian ecosystems to both extreme
wet events and droughts. Similarly the repeated frequency of heat extremes (van der Horst et al., 2019; van Gorsel et al., 2016;
Mitchell et al., 2014), offers important tests of underlying physiology assumptions. It is likely that insights gained from model
evaluation in Australia's environment of longer duration and high intensity climate extremes, including droughts, heatwaves
and floods, will help understand how Northern Hemisphere systems will respond to extremes, and in particular events that may
intensify in the future (IPCC, 2012). Finally, our study focused on Australia's carbon cycle, but the carbon and water cycle are



intimately linked and divergence between model simulations of the carbon cycle implies further work may also be needed to evaluate DGVM simulation of the hydrological cycle.

*Code and data availability.* All Eddy covariance data are available from http://www.ozflux.org.au/ (last access: 26 April 2017.) The Global Aboveground Biomass Carbon (version 1.0) dataset (Liu et al., 2015) is freely available from http://wald.anu.edu.au/global-biomass/ (last access 13 August 2020), and the GOSIF-GPP product (Li and Xiao, 2019) can be obtained from https://globalecology.unh.edu/data/GOSIF-GPP.html (last access 20 April 2020). Fire $CO_2$ emissions were provided by the Copernicus Atmosphere Monitoring Service Global Fire Assimilation System (CAMS GFAS; https://apps.ecmwf.int/datasets/data/cams-gfas/; last access 6 March 2021) and the Global Fire Emissions
Database version 4 (GFED4s) described in (van der Werf et al., 2017) are available from https://globalfiredata.org/pages/data/ (last access 11 July 2020). The MODIS/Terra Vegetation Continuous Fields dataset was provided by the NASA's Land Processes Distributed Active Archive Center (https://lpdaac.usgs.gov/products/mod44bv006/; last access 29 January 2021). The TRENDY v8 model output is available upon request (https://sites.exeter.ac.uk/trendy). All analysis scripts are accessible on https://github.com/lteckentrup/TRENDY_v8_Australia.

## Appendix A:  Forcing data and simulation protocol

*Climate forcing*

The modelling groups either chose the 0.5° CRU monthly historical forcing over 1901–2018 (Harris, 2019a) or the 0.5° CRU-JRA55 6–hourly historical forcing over 1901–2018 (Harris, 2019b) regridded to the CRU 0.5° grid. The variables temperature, downward solar radiation flux, specific humidity and precipitation in JRA-55 are aligned to temperature, cloud fraction, vapour pressure and precipitation in CRU TS (v4.03), respectively. Atmospheric pressure, downward longwave radiation
flux and the meridional and zonal components of wind speed are not modified. JRA-55 is used for the years from 1958 to 2018. For years between 1901 and 1957, random (but fixed) years from JRA-55 for 1958-1967 are used to fill.

*$CO_2$ concentration*

The atmospheric $CO_2$ concentration is derived from ice core $CO_2$ data merged with NOAA data from 1958 onwards. The forcing covers the years 1700–2018 incremented annually (Le Quéré et al., 2018). The data from March 1958 are monthly
averages from the Mauna Loa (MLO) and the South Pole Observatory (SPO) provided by NOAA's Earth System Research Laboratory http://www.esrl.noaa.gov/gmd/ccgg/trends/. When no SPO data are available (including prior to 1975), SPO is constructed from the 1976–2017 average MLO–SPO trend and average monthly departure. Data for 2016–2018 are preliminary values. Data prior to March 1958 are estimated with a cubic spline fit to ice core data from Joos and Spahni (2008).

*Land-use*

The land-use datasets are based on updated data from HYDE for the years 1960–2019, as well as the latest wood harvest data provided by the Food and Agriculture Organization (FAO). The land-use states and transitions are identical to the LUH2 v2h dataset (Hurtt et al., 2020) for the years from 1700 to 1949 (i.e. consistent with the input for CMIP6). Starting 1950, the land-use forcing is based on new inputs from HYDE, and new FAO data for the national wood harvest demands. This leads to differences in comparison to LUH2 v2h, primarily in Brazil.





In order to convert natural vegetation to managed pasture, the LUH2 guidelines suggested all natural vegetation is cleared. The conversion of natural vegetation to rangeland only requires the clearance of forests. However, each modelling group developed model-specific land-use forcings leading to inter-model differences. We show the change in agricultural landcover in appendix figure B6.

*Nitrogen deposition and fertilization*

For the years from 1850 to 2014, the TRENDY models were forced with the historical nitrogen deposition database and then transitioned to the future RCP8.5 nitrogen-deposition databases for the years from 2015 to 2018. Details are available at https://esgf-node.llnl.gov/search/input4mips/. Nitrogen fertiliser input datasets are available via the $N_2O$ Model Intercomparison Project ('NMIP'; Tian et al., 2018) NMIP assumes that the nitrogen input data remain unchanged in years 2015, 2016, 2017 and 2018. Manure is not included for the TRENDY simulations.

**A0.1   Simulation protocol**

The TRENDY models either simulated or prescribed vegetation cover. All models prescribed land-use change according to the LUH2 guidelines. All models used the atmospheric $CO_2$ concentration from the year 1700 (276.6ppm) and recycled the climate mean and variability from the years 1901–1920 for the spin-up. Land-use change was constant during the spin-up and the crops and pasture distribution was set constant to the 1700 set-up. Depending on the simulation, different drivers became

transient starting in 1701.

We use three simulations of the TRENDY experiment (see appendix figure B1; Friedlingstein et al., 2019): the run with transient atmospheric $CO_2$ concentration, and time-invariant climate forcing and land-use ('$CO_2$'); the run with transient atmospheric $CO_2$ concentration and climate forcing and time-invariant land-use ('$CO_2$ + CLIM'); and finally the run with transient atmospheric $CO_2$ concentration, climate forcing and land-use change ('$CO_2$ + CLIM + LUC').

*Spin-up*

For the spin-up, the atmospheric $CO_2$ concentration and landcover were set to the pre-industrial values of the year 1700. During the spin-up, the years from 1901 to 1920 from the climate forcing were recycled.

*CTRL*

After the spin-up, the atmospheric $CO_2$ concentration remained time-invariant for the years 1701–2018. The simulation

continued to recycle the 1901–1920 climate forcing and the land-use was prescribed to the distribution of the year 1700.

*$CO_2$*

After the spin-up, the $CO_2$ run used a transient atmospheric $CO_2$ forcing. The simulation continued to recycle the 1901–1920 climate forcing and the land-use was prescribed to the distribution of the year 1700 (as in the 'CTRL' run).

*$CO_2$ + CLIM*

After the spin-up, the $CO_2$ + CLIM run used a transient atmospheric $CO_2$ forcing. The simulation continued to recycle the 1901–1920 climate forcing until the year 1900 and the land-use was prescribed to the distribution of the year 1700. Starting from 1901, the climate forcing became transient.

*$CO_2$ + CLIM + LUC*



After the spin-up, the $CO_2$ + CLIM run used a transient atmospheric $CO_2$ and land-use forcing while the simulation contin-

ued to recycle the 1901–1920 climate forcing until the year 1900. Starting from 1901, the climate forcing became transient as
well.

The TRENDY protocol required all participating models to be in equilibrium for the CTRL simulation after the spin-up.
Further, all models had to simulate the net annual land flux as a carbon sink over 1990s and/or 2000s for the $CO_2$ + CLIM +
LUC run as constrained by global atmospheric and oceanic observations (Keeling and Manning, 2014). Lastly, the global net

annual land use flux (ELUC) had to be a carbon source over the 1990s (based on the $CO_2$ + CLIM and $CO_2$ + CLIM + LUC
simulation).





## Appendix B: Tables

**Table B1.** Information for the four fluxsites located in the Northern Australian Tropical Transect (NATT). This includes the mean annual precipitation (MAP), mean annual temperature (MAT) and the vegetation classification following the International Geosphere-Biosphere Programme (IGBP). The observed variables originate from eddy covariance data collected by the TERN-OzFlux facility. Further details of the site vegetation is available in Whitley et al. (2016).

|  | Howard Springs (AU-How) | Daly Uncleared (AU-DaS) | Dry River (AU-Dry) | Sturt Plains (AU-Stp) |
|---|---|---|---|---|
| Years | 2002–2018 (2002–2019) | 2008–2019 (2008-2018) | 2009–2018 (2008–2019) | 2009–2018 (2008–2018) |
| Latitude | -12.4952 | -14.1592 | -15.2588 | -17.1507 |
| Longitude | 131.15005 | 131.3881 | 132.3706 | 133.3502 |
| MAP [mm/ yr$^{-1}$] | 1662 | 1324 | 925 | 719 |
| MAT [°C] | 26.8 | 26.8 | 27.0 | 26.3 |
| Vegetation | Woody Savannas: Herbaceous and other understory systems, 30–60% forest canopy | Savannas: Herbaceous and other understory systems, 30–60% forest canopy | Savannas: Herbaceous and other understory systems, 30–60% forest canopy | Grasslands: Herbaceous types of cover. Tree and shrub cover is less than 10% |





**Table B2.** Years from 1901–2017 identified as El Niño, La Niña, positive Indian Ocean Dipole ('pIOD'), negative Indian Ocean Dipole event ('nIOD') and remaining years that are both ENSO and IOD neutral based on the analysis by NOAA (for the ENSO events) and Bureau of Meteorology (IOD events). Note that we define a year to start in July and end in June so that for example the 1991–1992 El Niño appears in this table as the 1991 El Niño. We follow Ummenhofer et al. (2011), the Bureau of Meteorology Australia (Bureau of Meteorology, Commonwealth of Australia, 2016) and the National Weather Service Climate Prediction Center (Climate Prediction Center Internet Team, 2021) for the identification of ENSO and IOD events.

| El Niño | | La Niña | | pIOD | | nIOD | | ENSO and IOD neutral | |
|---|---|---|---|---|---|---|---|---|---|
| 1902 | 1972 | 1903 | 1964 | 1902 | 1963 | 1906 | 1974 | 1901 | 1951 |
| 1905 | 1976 | 1906 | 1970 | 1913 | 1972 | 1909 | 1975 | 1904 | 1952 |
| 1911 | 1977 | 1909 | 1973 | 1919 | 1982 | 1915 | 1980 | 1907 | 1953 |
| 1914 | 1982 | 1910 | 1974 | 1923 | 1983 | 1916 | 1981 | 1908 | 1959 |
| 1923 | 1986 | 1916 | 1975 | 1926 | 1995 | 1917 | 1985 | 1912 | 1962 |
| 1925 | 1987 | 1917 | 1988 | 1935 | 1997 | 1930 | 1989 | 1920 | 1966 |
| 1930 | 1991 | 1922 | 1989 | 1944 | 2002 | 1933 | 1992 | 1921 | 1967 |
| 1940 | 1994 | 1924 | 1993 | 1945 | 2006 | 1942 | 1996 | 1927 | 1971 |
| 1941 | 1997 | 1928 | 1998 | 1946 | 2012 | 1958 | 1998 | 1929 | 1978 |
| 1957 | 2002 | 1933 | 1999 | 1957 | 2015 | 1960 | 2010 | 1931 | 1979 |
| 1963 | 2004 | 1938 | 2000 | 1961 | | 1964 | 2014 | 1932 | 1984 |
| 1965 | 2006 | 1942 | 2007 | | | | | 1934 | 1990 |
| 1968 | 2009 | 1949 | 2008 | | | | | 1936 | 2001 |
| 1969 | 2015 | 1950 | 2010 | | | | | 1937 | 2003 |
| | | 1954 | 2011 | | | | | 1939 | 2005 |
| | | 1955 | 2012 | | | | | 1943 | 2013 |
| | | 1956 | | | | | | 1947 | 2016 |
| | | | | | | | | 1948 | |





**Table B3.** Fire $CO_2$ emissions for Australia averaged over 2003–2018 and the Pearson correlation coefficients between the TRENDY models and the respective observation data. Bold numbers indicate a significant statistical relationship ((p-value < 0.05).

| Model | Fire $CO_2$ emissions (PgC yr-1) | R(CAMS-GFAS, model) | | R(GFED4s, model) | |
|---|---|---|---|---|---|
| | | Monthly | Annual | Monthly | Annual |
| CLASS-CTEM | $0.010 \pm 0.006$ | **0.68** | -0.49 | **0.78** | **-0.51** |
| CLM5.0 | $0.014 \pm 0.013$ | 0.12 | -0.67 | 0.08 | **-0.68** |
| ISBA-CTRIP | $0.029 \pm 0.021$ | **0.35** | 0.59 | **-0.41** | 0.47 |
| JSBACH | $0.012 \pm 0.007$ | **0.50** | -0.19 | **0.43** | -0.31 |
| LPX-Bern | $0.006 \pm 0.002$ | 0.05 | -0.26 | 0.04 | -0.3 |
| SDGVM | $0.048 \pm 0.011$ | 0.10 | 0.51 | 0.09 | 0.49 |
| VISIT | $0.026 \pm 0.010$ | **0.21** | -0.50 | **0.26** | -0.47 |
| CAMS-GFAS | $0.043 \pm 0.049$ | | | | |
| GFED4s | $0.030 \pm 0.029$ | | | | |



## Appendix B: Figures

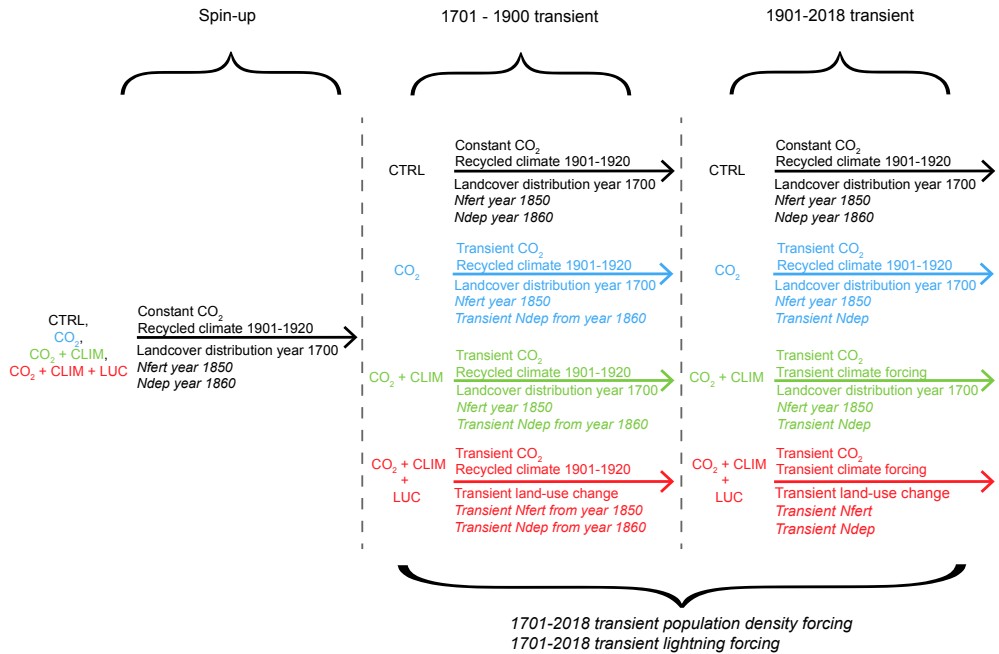

**Figure B1.** Simulations conducted by the TRENDY models (Friedlingstein et al., 2019). Italic forcing datasets are not used by all models.



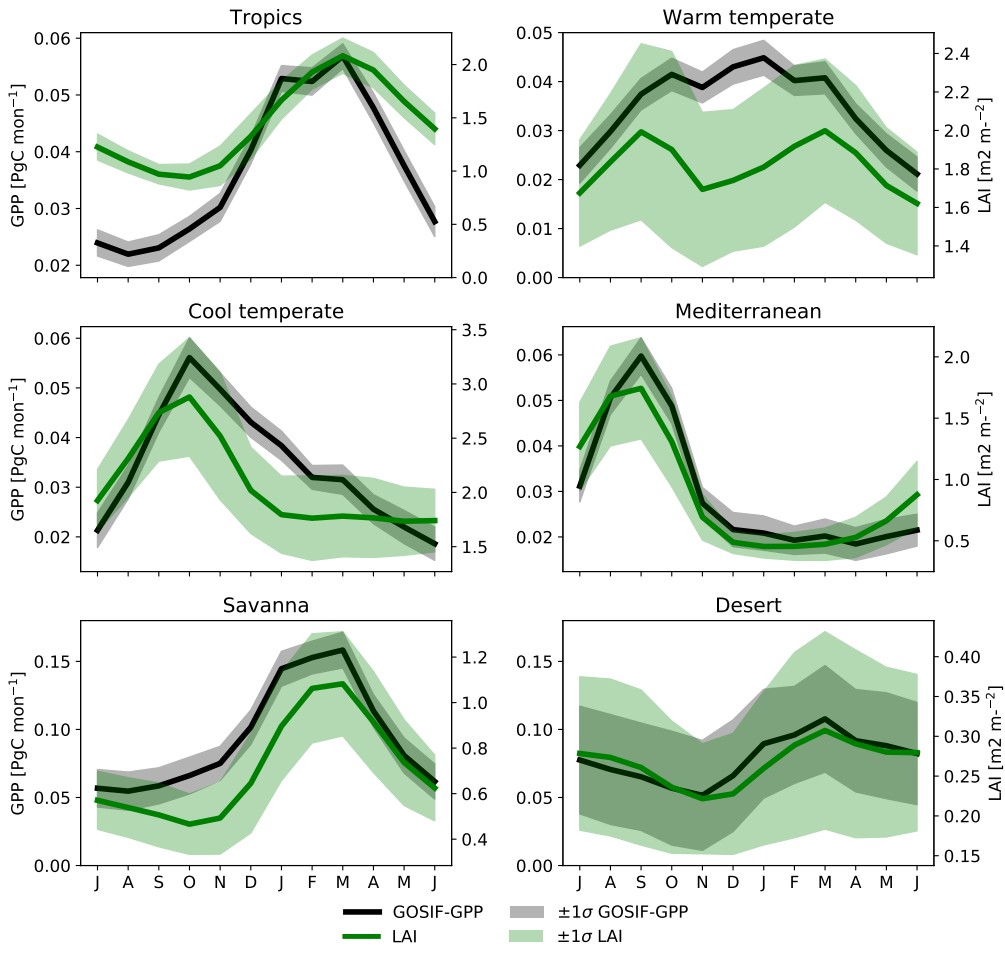

**Figure B2.** Seasonal cycle for satellite derived gross primary production (GOSIF-GPP; black) and satellite derived leaf area index data (LAI; Copernicus Global Land Service; green). Shaded areas indicate the according uncertainty (i.e. ± 1 standard deviation.)



**Figure B3.** Monthly temperature, precipitation and incoming shortwave radiation data that are observed (blue) and the corresponding data from the reanalysis product CRU-JRA (orange) for the four fluxsites located in the Northern Australian Tropical Transect (NATT): Howard Springs (AU-How), Daly Uncleared (AU-DaS), Dry River (AU-Dry) and Sturt Plains (AU-Stp). The figure titles display the Pearson correlation coefficient between observation and reanalysis ($\rho_{mon}$). The bottom panels show observed monthly NEE (black) and simulated NEE by the individual TRENDY models.




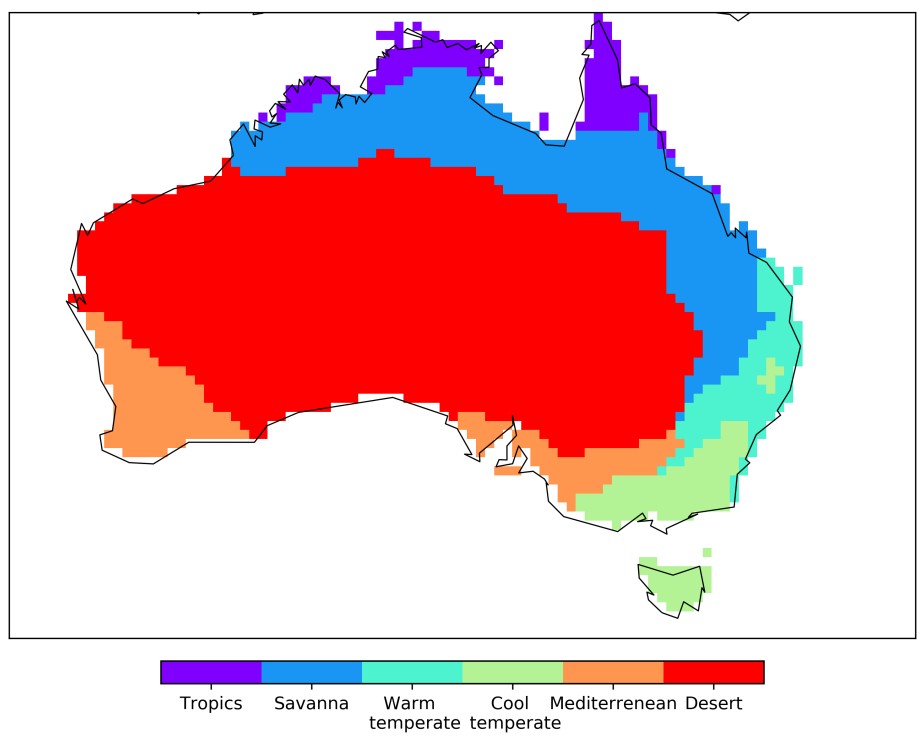

**Figure B4.** Different vegetation classes in Australia according to Haverd et al. (2012).



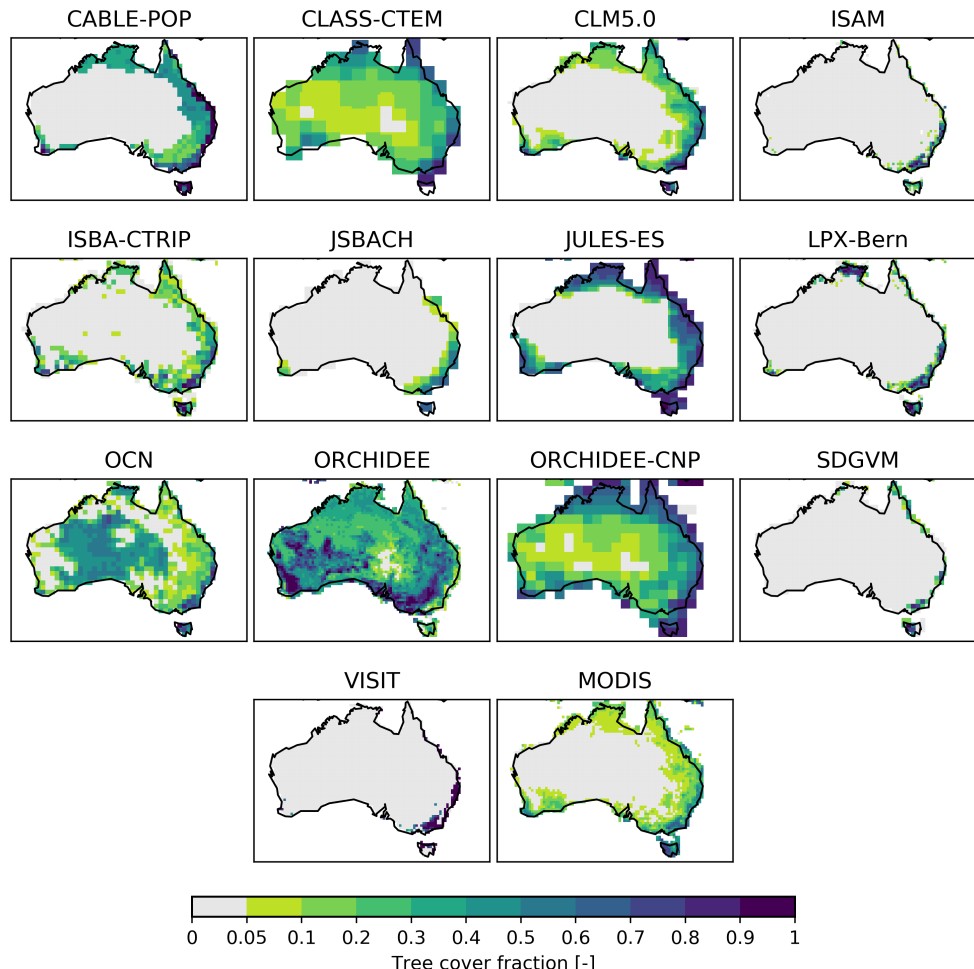

**Figure B5.** Landcover fraction of tree vegetation averaged over 2001–2018. Tree vegetation here shows the sum of tree, forest and shrub PFTs in the TRENDY models. The MODIS panel shows the tree fraction variable from the MODIS/Terra vegetation continuous fields dataset. White areas are missing values.



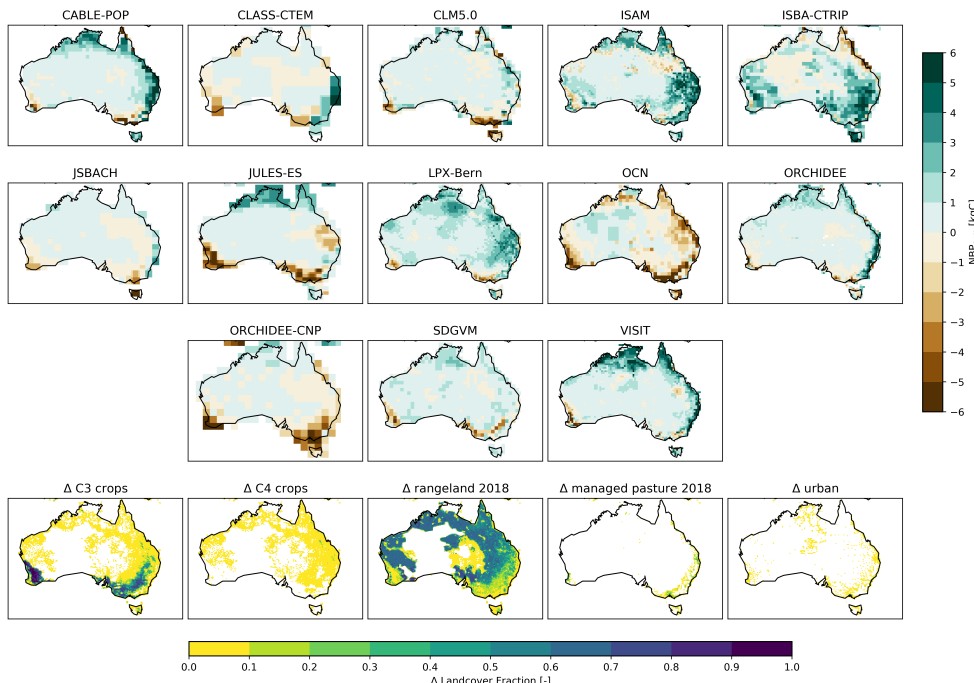

**Figure B6.** Cumulative NBP for all TRENDY models over 1901–2018. Land–use change based on Hurtt et al. (2020): Increase in C3 and C4 crops, rangeland, managed pasture and urban areas in 2018 compared to 1901.



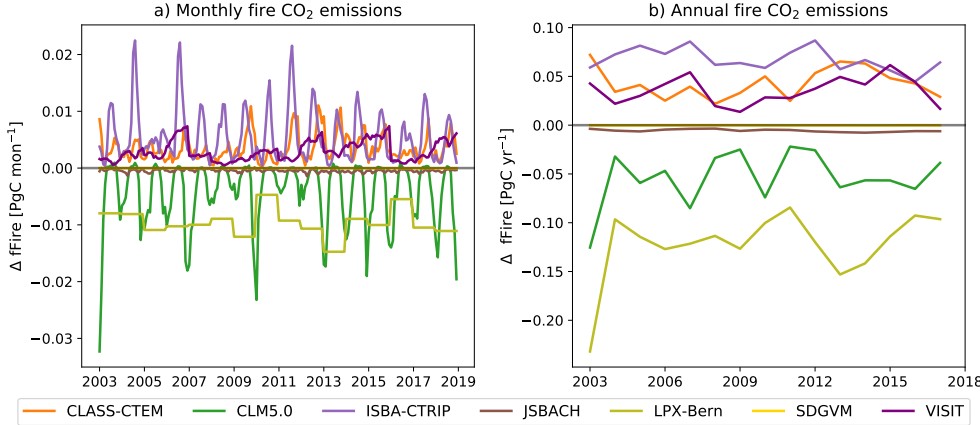

**Figure B7.** Difference between $CO_2$ + CLIM + LUC and $CO_2$ + CLIM simulations for monthly (a) and annual (b) fire $CO_2$ emissions of six TRENDY models.

*Author contributions.* LT, MDK and AJP designed the experimental analysis. LT wrote the code and analysed the TRENDY models. DG,
VH, AKJ, EJ, EK, SL, DL, PCM, JRM, JEMSN, and SZ conducted the model runs to produce the TRENDY data. LT wrote the first draft with
contributions from MDK and AJP. All authors contributed to the final manuscript. We thank Vladislav Bastrikov and Andrew J. Wiltshire for
providing model output as part of the TRENDY v8 ensemble.

*Competing interests.* The authors declare no conflict of interests. Correspondence and requests for materials should be addressed to LT
(l.teckentrup@unsw.edu.au)

*Acknowledgements.* LT, MDK and AJP acknowledge support Australian Research Council (ARC) Centre of Excellence for Climate Ex-
tremes (CE170100023). MDK and AJP acknowledge support from the ARC Discovery Grant (DP190101823). MDK was also supported
from the NSW Research Attraction and Acceleration Program. This work utilised data from the OzFlux network which is supported by the
Australian Terrestrial Ecosystem Research Network (TERN; http://www.tern.org.au).





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
