# Peer review of "Assessing the representation of the Australian carbon cycle in global vegetation models"

_Biogeosciences, 2021_

## Author Comment (AC1)

**Response to referee #1**

We thank the reviewer for taking the time to review our manuscript and provide constructive comments. Below we address the reviewer's comments point by point. We add our replies in italics and highlight suggested modifications in the manuscript in red.

We further would like to acknowledge that while revising the manuscript, we noticed two minor errors in the original submission, which we have now fixed:

1.  Figure 1d): For two models (ISBA-CTRIP and VISIT) the monthly CSoil pool was shown. We corrected the figure and now show the annual average for both models.

2.  Figure 7c) and Table B3: We accidentally chose the wrong time period for the annual fire emissions and updated the figures with the corrected time period (2003-2018 for the TRENDY models and the observation data). We accordingly corrected Table B3. While the annual values for fire emissions and the associated uncertainty almost stay the same, the correlation coefficients between the TRENDY models and the observed data changed so that now all models are positively correlated with either of the two observation datasets on both timescales. We changed the text accordingly to:

    *On monthly timescales, four TRENDY models capture some features in the variability in the satellite derived observations with either weak (ISBA-CTRIP and VISIT: both datasets; JSBACH: GFED4s), moderate (JSBACH: CAMS-GFAS; CLASS-CTEM: CAMS-GFAS) or high (CLASS-CTEM: GFED4s) significant correlation coefficients. The remaining models do not show a significant relationship to either of the datasets. Aggregated to annual values, the TRENDY models generally underestimated the fire $CO_2$ emissions and did not always capture the variability in, or timing of, extreme fire years (see fig. 7c). CLASS-CTEM, JSBACH and ISBA-CTRIP captured some features of the variability in the satellite derived observations. CLASS-CTEM is moderately correlated with both datasets ($r>0.5$), ISBA-CTRIP shows a significant moderate correlation with the GFED4s dataset only and JSBACH is highly correlated with both datasets ($r>0.7$; see table B3). The remaining models are not significantly linked the satellite derived observations.*

**General comments**

Using the term "DGVM" is somewhat misleading because some of the models use prescribed PFT fractions. Explicitly identifying which models are truly DGVMs and which are not at the beginning instead of in the Future Directions section at the end might help a reader while digesting the figures.

*We agree with the reviewer. However, we followed the TRENDY nomenclature which states that the TRENDY project is a 'consortium of Dynamic Global Vegetation Model (DGVM) groups'. To clarify, we now list the models that simulate vegetation dynamically as opposed to those prescribing landcover in the section 2.1 Models and simulations (instead of the Future Directions). We hope this resolves the reviewer's concern.*

Some additional analyses that bring the separate sections together will help disentangle the results. Most importantly, more detailed analyses of the effect of PFT fraction might allow for

more informative conclusions. For instance, one conclusion is that differences in PFT fraction are largely responsible for differences in carbon storage and turnover time. Showing each model's NBP difference from the multi-model mean (and carbon storage and residence time) vs. the fraction of each PFT, either for every year or averaged over a number of years, would help understand the role of vegetation distribution on differences in the simulated response variables of interest.

*We thank the reviewer for the suggestion. Given all models use different PFTs, we grouped the PFTs of the individual models into six common vegetation groups, i.e., evergreen trees, deciduous trees, C3 grasses (C3G), C4 grasses (C4G), C3 agriculture (C3 Crop) and C4 agriculture (C4 Crop). We exclude grid points where the landcover fraction of the vegetation group is less than 5%.*
*We looked at NBP (sum over 1901-2018) – ensemble mean vs the initial landcover fraction (1901-1930):*

[Figure]

*Figure 1: NBP sum over 1901-2018 minus the ensemble mean against the initial landcover fraction (1901-1930) for six different vegetation groups (evergreen trees, deciduous trees, C3 grasses (C3G), C4 grasses (C4G), C3 agriculture (C3 Crop) and C4 agriculture (C4 Crop). The lines show the linear regression lines.*

*And vs the landcover fraction at the end of simulation (1989-2018):*

[Figure]

*Figure 2: NBP sum over 1901-2018 minus the ensemble mean against the landcover fraction at the end of the simulation (1989-2018) for six different vegetation groups (evergreen trees, deciduous trees, C3 grasses (C3G), C4 grasses (C4G), C3 agriculture (C3 Crop) and C4 agriculture (C4 Crop). The lines show the linear regression lines.*

*In our opinion, unfortunately neither of the figures adds value to our manuscript. It is hard to see whether the deviation from the ensemble mean changes with the landcover.*

*We further looked at CVeg – ensemble mean (averaged over 1989-2018) vs the initial landcover fraction (1901-1930):*

[Figure]

*Figure 3: Average carbon stored in vegetation (CVeg; 1989-2018) minus the ensemble mean against the initial landcover fraction (1901-1930) for six different vegetation groups (evergreen trees, deciduous trees, C3 grasses (C3G), C4 grasses (C4G), C3 agriculture (C3 Crop) and C4 agriculture (C4 Crop). The lines show the linear regression lines.*

*And CVeg – ensemble mean (averaged over 1989-2018) vs landcover fraction (1989-2018):*

[Figure]

*Figure 4: Average carbon stored in vegetation (CVeg; 1989-2018) minus the ensemble mean against the landcover fraction at the end of the simulation (1989-2018) for six different vegetation groups (evergreen trees, deciduous trees, C3 grasses (C3G), C4 grasses (C4G), C3 agriculture (C3 Crop) and C4 agriculture (C4 Crop). The lines show the linear regression lines.*

*As opposed to the figure with NBP, here the vegetation groups deciduous trees, C3 grasses, C3 crops and C4 grasses appear to decrease with increasing landcover fraction. However, this is likely an artefact given that for most models there are more pixels with a landcover fraction with <30%.*

*Lastly, we explored the CSoil pools (CSoil – ensemble mean, averaged over 1989-2018) against the initial landcover fraction (1901-1930):*

[Figure]

*Figure 5: Average carbon stored in soil (CSoil; 1989-2018) minus the ensemble mean against the initial landcover fraction (1901-1930) for six different vegetation groups (evergreen trees, deciduous trees, C3 grasses (C3G), C4 grasses (C4G), C3 agriculture (C3 Crop) and C4 agriculture (C4 Crop). The lines show the linear regression lines.*

*And CSoil – ensemble mean vs landcover fraction (1989-2018):*

[Figure]

*Figure 6: Average carbon stored in soil (CSoil; 1989-2018) minus the ensemble mean against the landcover fraction at the end of the simulation (1989-2018) for six different vegetation groups (evergreen trees, deciduous trees, C3 grasses (C3G), C4 grasses (C4G), C3 agriculture (C3 Crop) and C4 agriculture (C4 Crop). The lines show the linear regression lines.*

*Similar to the figure NBP vs landcover fraction, we do not think the CSoil figure would add value to our manuscript because it is hard to draw any conclusions. We have therefore opted to keep our original figure presentation in the manuscript.*

The discussion of inter-annual variability in precipitation would also benefit from assessing the model differences in terms of PFT fraction. This is alluded to in the Discussion but could be explicitly evaluated.

*Similar to above, we took the average over the years 1989-2018 for precipitation (CRUJRA reanalysis) and landcover fraction for the six different vegetation groups. We first compared the average precipitation against the landcover fraction:*

[Figure]

*Figure 7: Average annual precipitation (PPT; 1989-2018) against the landcover fraction at the end of the simulation (1989-2018) for six different vegetation groups (evergreen trees, deciduous trees, C3 grasses (C3G), C4 grasses (C4G), C3 agriculture (C3 Crop) and C4 agriculture (C4 Crop). The lines show the linear regression lines.*

*And further compare the standard deviation (as a measure of interannual variability) against the landcover fraction:*

[Figure]

*Figure 8: Standard deviation of annual precipitation (PPT) over 1989-2018 against the landcover fraction at the end of the simulation (1989-2018) for six different vegetation groups (evergreen trees, deciduous trees, C3 grasses (C3G), C4 grasses (C4G), C3 agriculture (C3 Crop) and C4 agriculture (C4 Crop). The lines show the linear regression lines.*

*Again, as shown in our earlier figures, the same issues apply. Furthermore, we don't necessarily think there is a mechanistic reason to assume a direct link between IAV of precipitation and land cover. This is because although IAV of rainfall would drive growth/mortality dynamics, these processes will be lagged and consequently the relationships are as seen, less clear.*

Finally, analysis of differences in burned area relative to PFT fraction would be helpful.

*Similar to above, we use the 1989-2018 average for the landcover fraction for the six different vegetation groups and the burned area fraction. Only six models provide burned area output.*

[Figure]

*Figure 9: Average burned area fraction (BA; 1989-2018) minus the ensemble mean against the landcover fraction at the end of the simulation (1989-2018) for six different vegetation groups (evergreen trees, deciduous trees, C3 grasses (C3G), C4 grasses (C4G), C3 agriculture (C3 Crop) and C4 agriculture (C4 Crop). The lines show the linear regression lines.*

*For two groups (evergreen trees and C3 grasses) it looks like the deviation from the ensemble mean might decrease with increasing landcover fraction. However, it is obvious there are more data points in the 0-30% range for landcover fraction than for values greater than 30%. The other vegetation groups do not offer interesting insights in our opinion, and we therefore choose to not include this figure in the revised manuscript.*

Plotting annual average NBP vs. annual average area burned (or emissions), with error bars, could help understand the role of fire on differences in NBP.

*We thank the reviewer for the suggestion. We explored this in some detail, including plotting annual average burned area and fire CO2 emissions against average NBP Unfortunately, the results do not provide significant additional insights into the role of fire on NBP. We will try to explore this more in the future, but for this manuscript we have not added in further details.*

[Figure]

*Figure 10: NBP against burned area fraction (BAF; upper panel) and NBP against fire CO2 emissions (fFire; lower panel). All variables are averaged over the time period 1989-2018.*

It might help if the Results and Discussion presented the variables contributing to differences among models in the same order. PFT fraction is presented last in the Results, but addressed first in the Discussion.

*We thank the reviewer for this suggestion, and we revised the structure of the discussion as suggested.*

The colors for each model are hard to distinguish in the time series plots. Maybe including different line types in addition to colors would help.

*We agree and have updated the figures accordingly.*

**Specific Comments:**

Lines 6-7: Is there a word missing in this sentence?

*We changed the text to*
*The TRENDY models simulated differing magnitudes of NBP on inter-annual timescales, and these differences resulted in significant differences in long-term vegetation carbon accumulation (-4.7-9.5 PgC).*

Lines 162-163: Plotting NBP anomaly vs ensemble spread would help make the point that years with extremely low or high NPB have large uncertainty more clear.

*We thank the reviewer for the suggestion. However, as NBP varies around zero for all models, a 30-year-average will be close to zero, resulting in anomalies close to the actual NBP. In consequence, it is hard to see a difference between total NBP and NBP anomalies. We have shown this below (the first figure is the original and the second figure is the figure recreated with the anomalies). Consequently, as the anomaly does not provide any further insight, we have kept our original presentation in the manuscript.*

[Figure]

*Fig 11: Original figure 1 in manuscript.*

[Figure]

*Figure 12: NBP anomalies instead of total NBP.*

Figure 2: Plotting the differences between CO2-only and the other two experiments (similar to Figure B7, except summed over the simulations) for each model would make it easier to see the effects of climate and LUC.

*We thank the reviewer for their suggestion. We agree that this is a possible way to present the attribution. However, we would like to keep our presentation to show the combined effect of the different factors. This allows us to explore possible synergy effects between the three drivers CO2, climate, and land-use change.*

Figure 8. Showing observed land cover fraction would be helpful.

*We agree with the reviewer. While different datasets exist that describe the landcover in Australia, to our knowledge no dataset exists that describes both the extent of vegetation types as well as their fraction. The closest we could find was the MODIS/Terra vegetation continuous fields dataset which only includes the three vegetation groups (Tree, non-tree, not vegetated; see fig. 9). However, the MODIS product isn't trained for the specifics of the Australian environment (e.g., savanna ecosystems) and we have concerns that presenting it as an objective truth would not help the analysis.*

Line 297: Sentence "processes landcover/land use change" is missing "of".

*Thank you, we changed the text accordingly.*

Line 334: Change "an nitrogen cycle", to "a nitrogen cycle"

*Thank you, we changed the text accordingly.*

Line 359-360: These relationships, especially, between PFT fraction and residence time, could be checked relatively easily as suggested in my general comments.

*Thank you. As described above, we explored the different suggestions made and have come to the conclusion that unfortunately none of them would add value to our manuscript. One of the difficulties we were facing was that it is hard to compress all the information required into a comprehensible figure. Combining different vegetation groups for 13 models in form of scatter plots makes it hard to draw any conclusions. We alternatively discussed only showing the linear regression lines, however, while most models show significant linear relationships, the adjusted R-squared values are low and RMSE values high for most models for all variables. Therefore, we concluded the linear regression lines are not representative of the data but are too noisy to derive a meaningful linear relationship instead.*

Line 369: Change "was less important factor" to "was a less important factor"

*Thank you, we changed the text accordingly.*

Lines 441-456: Why not compare the parameterizations these 13 models used? That, in conjunction with the additional plots of NBP, carbon storage, and residence time by PFT fractions could allow for more comprehensive and informative conclusions as to why the models differ.

*It is difficult to compare the parameterisations directly outside of the individual schemes as the parameterisations are woven with the parameters chosen by each model, as well as issues like the timescales (i.e., changes in cover type and so dominant parameters) over which each scheme responds to the meteorological forcing. We have therefore not compared the 13 models because we do not think this would actually lead to insight. In the future direction we discuss the importance of the parametrisations and highlight that further work linking model parameters to emergent trait databases and field data (e.g., AusTraits database, Bloomfield et al., 2019, Togashi et al., 2015) is an important step to improve the model performance over Australia (see l. 446 – 452).*

---

## Author Comment (AC2)

**Response to referee #2**

We thank the reviewer for taking the time to review our manuscript and provide constructive comments. Below we address the reviewer's comments point by point. We add our replies in italics and highlight suggested modifications in the manuscript in red.

We further would like to acknowledge that while revising the manuscript, we noticed two minor errors in the original submission, which we have now fixed:

1. Figure 1d): For two models (ISBA-CTRIP and VISIT) the monthly CSoil pool was shown. We corrected the figure and now show the annual average for both models.

2. Figure 7c) and Table B3: We accidentally chose the wrong time period for the annual fire emissions and updated the figures with the corrected time period (2003-2018 for the TRENDY models and the observation data). We accordingly corrected Table B3. While the annual values for fire emissions and the associated uncertainty almost stay the same, the correlation coefficients between the TRENDY models and the observed data changed so that now all models are positively correlated with either of the two observation datasets on both timescales. We changed the text accordingly to:

   *On monthly timescales, four TRENDY models capture some features in the variability in the satellite derived observations with either weak (ISBA-CTRIP and VISIT: both datasets; JSBACH: GFED4s), moderate (JSBACH: CAMS-GFAS; CLASS-CTEM: CAMS-GFAS) or high (CLASS-CTEM: GFED4s) significant correlation coefficients. The remaining models do not show a significant relationship to either of the datasets. Aggregated to annual values, the TRENDY models generally underestimated the fire CO$_2$ emissions and did not always capture the variability in, or timing of, extreme fire years (see fig. 7c). CLASS-CTEM, JSBACH and ISBA-CTRIP captured some features of the variability in the satellite derived observations. CLASS-CTEM is moderately correlated with both datasets (r>0.5), ISBA-CTRIP shows a significant moderate correlation with the GFED4s dataset only and JSBACH is highly correlated with both datasets (r>0.7; see table B3). The remaining models are not significantly linked the satellite derived observations.*

**General comments**

This manuscript analyzes how the Australian carbon cycle is simulated by 13 vegetation models that exbibit large differences in their outputs and behavior. It's an interesting and comprehensive analysis, looking at factors such as carbon residence time, land cover differences, and fire.

*We thank the reviewer for the positive assessment of our work.*

There are some minor weaknesses. The text is unclear in some spots, and some of the figures could be improved or re-thought.

*We thank the reviewer for their input. As described in the response to reviewer 1, we tried to incorporate suggestions made but none of them added insight beyond the original figures presented so we have kept the original presentation. It is of course challenging with 13 models and multiple carbon axes of interest, but by separating by experiment (CO2, climate), examining as a function of time, landcover, etc, we have elicited important insights into model behaviour/skill across the Australian continent. Importantly, this comprehensive assessment was currently lacking. Finally, we have improved the highlighted sentences as suggested (see below).*

Finally, I didn't see anything about data or code availability; this is critical for transparency and reproducibility.

*We're unclear what we have missed here and are happy to adjust the text as the Reviewer deems necessary. Our data availability section listed a link to our github repository with all the analysis code and we put all the links to the datasets used to evaluate the models.*

In summary, this is a well-done and interesting analysis that needs moderate revisions in many areas for clarity and concision.

**Specific comments**

1. Lines 18-20: somewhat confusing

   *We thank the reviewer for the comment and rephrased the text to:*
   *In addition, we find that differences in the timing of simulated phenology and fire dynamics are associated with differences in simulated/prescribed vegetation cover and process representation. We further find model disagreement in simulated vegetation carbon, phenology and apparent carbon residence time, indicating that the models have different types and coverage of vegetation across Australia (whether prescribed or emergent).*

2. 54 and 465: perhaps start new paragraph

   *We updated the manuscript accordingly.*

3. 80: what is NATT? Not defined yet

   *We thank the reviewer and updated the manuscript:*
   *We remapped all model outputs and satellite datasets (see below) to a common 0.5°C grid using first order conservative regridding (except for the comparison with the data over the North Australian Tropical Transect (NATT)).*

4. 177-179: unnecessary? Perhaps move to figure caption

   *We thank the reviewer for the suggestion. However, as this is a description of results, we feel it does belong in the results section rather than in the figure caption.*

5.  Figure 1: why does panel a show the ensemble mean and spread but other panels show individual models? Perhaps add green mean to panels b-d?

    *Since all models vary around zero for NBP, it is hard to distinguish individual models over a time period of 118 years when each model is shown. We therefore included the spread to show the uncertainty associated with simulated NBP. In contrast, the remaining three variables have lower interannual variability and showing individual models allows to see the different trajectories they take over time. In summary, while we're not advocates for the use of the ensemble mean, we do so in panel a as we think it adds value to the interpretation for the reader.*

6.  Not sure how useful figure 7a is

    *We aimed to explore the possible link between ENSO and fire regimes in Australia with panel a (as described in l. 247-253). We do agree that if we were to delete a panel, we would delete panel (a) but on balance we have decided to leave it in. We would be open to further advice from the editor on this of course.*

7.  Figures 8 and 9 are interesting!

    *We thank the reviewer for their positive assessment.*